# Harnessing the Power of Choices in Decision Tree Learning

**Guy Blanc***
*Stanford*
gblanc@stanford.edu

**Jane Lange***
*MIT*
jlange@mit.edu

**Chirag Pabbaraju***
*Stanford*
cpabbara@stanford.edu

**Colin Sullivan***
*Stanford*
colins26@stanford.edu

**Li-Yang Tan***
*Stanford*
lytan@stanford.edu

**Mo Tiwari***
*Stanford*
motiwari@stanford.edu

## Abstract

We propose a simple generalization of standard and empirically successful decision tree learning algorithms such as ID3, C4.5, and CART. These algorithms, which have been central to machine learning for decades, are greedy in nature: they grow a decision tree by iteratively splitting on the best attribute. Our algorithm, Top-$k$, considers the $k$ best attributes as possible splits instead of just the single best attribute.We demonstrate, theoretically and empirically, the power of this simple generalization. We first prove a *greediness hierarchy theorem* showing that for every $k \in \mathbb{N}$, Top-$(k + 1)$ can be dramatically more powerful than Top-$k$: there are data distributions for which the former achieves accuracy $1 - \varepsilon$, whereas the latter only achieves accuracy $\frac{1}{2} + \varepsilon$. We then show, through extensive experiments, that Top-$k$ outperforms the two main approaches to decision tree learning: classic greedy algorithms and more recent "optimal decision tree" algorithms. On one hand, Top-$k$ consistently enjoys significant accuracy gains over greedy algorithms across a wide range of benchmarks. On the other hand, Top-$k$ is markedly more scalable than optimal decision tree algorithms and is able to handle dataset and feature set sizes that remain far beyond the reach of these algorithms. The code to reproduce our results is available at: `https://github.com/SullivanC19/pydl8.5-topk`.

## 1   Introduction

Decision trees are a fundamental workhorse in machine learning. Their logical and hierarchical structure makes them easy to understand and their predictions easy to explain. Decision trees are therefore the most canonical example of an interpretable model: in his influential survey [Bre01b], Breiman writes "On interpretability, trees rate an A+"; much more recently, the survey [RCC+22] lists decision tree optimization as the very first of 10 grand challenges for the field of interpretable machine learning. Decision trees are also central to modern ensemble methods such as random forests [Bre01a] and XGBoost [CG16], which achieve state-of-the-art accuracy for a wide range of tasks.

Greedy algorithms such as ID3 [Qui86], C4.5 [Qui93], and CART [BFSO84] have long been the standard approach to decision tree learning. These algorithms build a decision tree from labeled data in a top-down manner, growing the tree by iteratively splitting on the "best" attribute as measured with respect to a certain heuristic function (e.g., information gain). Owing to their simplicity, these

---

*Authors ordered alphabetically.

37th Conference on Neural Information Processing Systems (NeurIPS 2023).

algorithms are highly efficient and scale gracefully to handle massive datasets and feature set sizes, and they continue to be widely employed in practice and enjoy significant empirical success. For the same reasons, these algorithms are also part of the standard curriculum in introductory machine learning and data science courses.

The trees produced by these greedy algorithms are often reasonably accurate, but can nevertheless be suboptimal. There has therefore been a separate line of work, which we review in Section 2, on algorithms that optimize for accuracy and seek to produce *optimally* accurate decision trees. These algorithms employ a variety of optimization techniques (including dynamic programming, integer programming, and SAT solvers) and are completely different from the simple greedy algorithms discussed above. Since the problem of finding an optimal decision tree has long been known to be NP-hard [HR76], *any* algorithm must suffer from the inherent combinatorial explosion when the instance size becomes sufficiently large (unless P=NP). Therefore, while this line of work has made great strides in improving the scalability of algorithms for optimal decision trees, dataset and feature set sizes in the high hundreds and thousands remain out of reach.

This state of affairs raises a natural question:

> *Can we design decision tree learning algorithms that improve significantly on the accuracy of classic greedy algorithms and yet inherit their simplicity and scalability?*

In this work, we propose a new approach and make a case that provides a strong affirmative answer to the question above. Our work also opens up several new avenues for exploration in both the theory and practice of decision tree learning.

## 1.1 Our contributions

### 1.1.1 Top-$k$: a simple and effective generalization of classic greedy decision tree algorithms

We introduce an easily interpretable greediness parameter to the class of all greedy decision tree algorithms, a broad class that encompasses ID3, C4.5, and CART. This parameter, $k$, represents the number of features that the algorithm considers as candidate splits at each step. Setting $k = 1$ recovers the fully greedy classical approaches, and increasing $k$ allows the practitioner to produce more accurate trees at the cost of only a mild training slowdown. The focus of our work is on the regime where $k$ is a small constant—preserving the efficiency and scalability of greedy algorithms is a primary objective of our work—although we mention here that by setting $k$ to be the dimension $d$, our algorithm produces an optimal tree. Our overall framework can thus be viewed as interpolating between greedy algorithms at one extreme and "optimal decision tree" algorithms at the other, precisely the two main and previously disparate approaches to decision tree learning discussed above.

We will now describe our framework. A *feature scoring function* $\mathcal{H}$ takes as input a dataset over $d$ binary features and a specific feature $i \in [d]$, and returns a value quantifying the "desirability" of this feature as the root of the tree. The greedy algorithm corresponding to $\mathcal{H}$ selects as the root of the tree the feature that has the largest score under $\mathcal{H}$; our generalization will instead consider the $k$ features with the $k$ highest scores.

**Definition 1** (Feature scoring function). *A feature scoring function $\mathcal{H}$ takes as input a labeled dataset $S$ over a $d$-dimensional feature space, a feature $i \in [d]$, and returns a score $\nu_i \in [0, 1]$.*

See Section 3.1 for a discussion of the feature scoring functions that correspond to standard greedy algorithms ID3, C4.5, and CART. Pseudocode for Top-$k$ is provided in Figure 1. We note that from the perspective of interpretability, the trained model looks the same regardless of what $k$ is. During training, the algorithm considers more splits, but only one split is eventually used at each node.

### 1.1.2 Theoretical results on the power of Top-$k$

The search space of Top-$(k + 1)$ is larger than that of Top-$k$, and therefore its training accuracy is certainly at least as high. The first question we consider is: is the test accuracy of Top-$(k + 1)$ only marginally better than that of Top-$k$, or are there examples of data distributions for which even a single additional choice provably leads to huge gains in test accuracy? Our first main theoretical result is a sharp *greediness hierarchy theorem*, showing that this parameter can have dramatic impacts on accuracy, thereby illustrating its power:

Top-$k(\mathcal{H}, S, h)$:

> **Given:** A feature scoring function $\mathcal{H}$, a labeled sample set $S$ over $d$ dimensions, and depth budget $h$.
>
> **Output:** Decision tree of depth $h$ that approximately fits $S$.
>
> 1. If $h = 0$, or if every point in $S$ has the same label, return the constant function with the best accuracy w.r.t. $S$.
>
> 2. Otherwise, let $\mathcal{I} \subseteq [d]$ be the set of $k$ coordinates maximizing $\mathcal{H}(S, i)$.
>
> 3. For each $i \in \mathcal{I}$, let $T_i$ be the tree with
>
> $$\text{Root} = x_i$$
> $$\text{Left subtree} = \text{Top-}k(\mathcal{H}, S_{x_i=0}, h-1)$$
> $$\text{Right subtree} = \text{Top-}k(\mathcal{H}, S_{x_i=1}, h-1),$$
>
> where $S_{x_i=b}$ is the subset of points in $S$ where $x_i = b$.
>
> 4. Return the $T_i$ with maximal accuracy with respect to $S$ among all choices of $i \in \mathcal{I}$.

Figure 1: The Top-$k$ algorithm. It can be instantiated with any feature scoring function $\mathcal{H}$, and when $k = 1$, recovers standard greedy algorithms such as ID3, C4.5, and CART.

**Theorem 1** (Greediness hierarchy theorem). *For every $\varepsilon > 0$, $k, h \in \mathbb{N}$, there is a data distribution $\mathcal{D}$ and sample size $n$ for which, with high probability over a random sample $\boldsymbol{S} \sim \mathcal{D}^n$, Top-$(k+1)$ achieves at least $1 - \varepsilon$ accuracy with a depth budget of $h$, but Top-$k$ achieves at most $\frac{1}{2} + \varepsilon$ accuracy with a depth budget of $h$.*

All of our theoretical results, Theorems 1 to 3, hold whenever the scoring function is an *impurity-based heuristic*. This broad class includes the most popular scoring functions (see Section 3.1 for more details). Theorem 1 is a special case of a more general result that we show: for all $k < K$, there are data distributions on which Top-$K$ achieves maximal accuracy gains over Top-$k$, even if Top-$k$ is allowed a larger depth budget:

**Theorem 2** (Generalization of Theorem 1). *For every $\varepsilon > 0$, $k, K, h \in \mathbb{N}$ where $k < K$, there is a data distribution $\mathcal{D}$ and sample size $n$ for which, with high probability over a random sample $\boldsymbol{S} \sim \mathcal{D}^n$, Top-$K$ achieves at least $1 - \varepsilon$ accuracy with a depth budget of $h$, but Top-$k$ achieves at most $\frac{1}{2} + \varepsilon$ accuracy even with a depth budget of $h + (K - k - 1)$.*

The proof of Theorem 2 is simple and highlights the theoretical power of choices. One downside, though, is that it is based on data distributions that are admittedly somewhat unnatural: the labeling function has embedded within it a function that is the XOR of certain features, and real-world datasets are unlikely to exhibit such adversarial structure. To address this, we further prove that the power of choices is evident even for *monotone* data distributions. We defer the definition of monotone data distributions to Section 4.2.

**Theorem 3** (Greediness hierarchy theorem for monotone data distributions). *For every $\varepsilon > 0$, depth budget $h$, $K$ between $\tilde{\Omega}(h)$ and $\tilde{O}(h^2)$ and $k \leq K - h$, there is a monotone data distribution $\mathcal{D}$ and sample size $n$ for which, with high probability over a random sample $\boldsymbol{S} \sim \mathcal{D}^n$, Top-$K$ achieves at least $1 - \varepsilon$ accuracy with a depth budget of $h$, but Top-$k$ achieves at most $\frac{1}{2} + \varepsilon$ accuracy with a depth budget of $h$.*

Many real-world data distributions are monotone in nature, and relatedly, they are a common assumption and the subject of intensive study in learning theory. Most relevant to this paper, recent theoretical work has identified monotone data distributions as a broad and natural class for which classical greedy decision tree algorithms (i.e., Top-1) provably succeed [BLT20b, BLT20a]. Theorem 3 shows that even within this class, increasing the greediness parameter can lead to dramatic gains in accuracy. Compared to Theorem 2, the proof of Theorem 3 is more technical and involves the use of concepts from the Fourier analysis of boolean functions [O'D14].

We note that a weaker version of Theorem 3 is implicit in prior work: combining [BLT20b, Theorem 7b] and [BLQT21b, Theorem 2] yields the special case of Theorem 3 where $K = O(h^2)$ and $k = 1$. Theorem 3 is a significant strengthening as it allows for $k > 1$ and much smaller $K - k$.

### 1.1.3 Experimental results on the power of Top-$k$

We provide extensive empirical validation of the effectiveness of Top-$k$ when trained on on real-world datasets, and provide an in-depth comparison with both standard greedy algorithms as well as optimal decision tree algorithms.

We first compare the performance of Top-$k$ for $k = 1, 2, 3, 4, 8, 12, 16$ (Figure 2), and find that increasing $k$ does indeed provide a significant increase in test accuracy—in some cases, Top-8 already achieves accuracy comparable to the test accuracy attained by DL8.5 [ANS20], an optimal decision tree algorithm. We further show, in Figures 3 and 6, that Top-$k$ inherits the efficiency of popular greedy algorithms and scales much better than the state-of-the-art optimal decision tree algorithms MurTree and GOSDT [LZH+20].

Taken as a whole, our experiments demonstrate that Top-$k$ provides a useful middle ground between greedy and optimal decision tree algorithms: it is significantly more accurate than greedy algorithms, but still fast enough to be practical on reasonably large datasets. See Section 5 for an in-depth discussion of our experiments. Finally, we emphasize the benefits afforded by the simplicity of Top-$k$. Standard greedy algorithms (i.e. Top-1) are widely employed and easily accessible. Introducing the parameter $k$ requires modifying only a tiny amount of source code and gives the practitioner a new lever to control. Our experiments and theoretical results demonstrate the utility of this simple lever.

## 2 Related work

**Provable guarantees and limitations of greedy decision tree algorithms.** A long and fruitful line of work seeks to develop a rigorous understanding of the performances of greedy decision tree learning algorithms such as ID3, C4.5, and CART and to place their empirical success on firm theoretical footing [KM96, Kea96, DKM96, BDM19, BDM20, BLT20b, BLT20a, BLQT21a]. These works identify feature and distributional assumptions under which these algorithms provably succeed; they also highlight the *limitations* of these algorithms by pointing out settings in which they provably fail. Our work complements this line of work by showing, theoretically and empirically, how these algorithms can be further improved with a simple new parameter while preserving their efficiency and scalability.

**The work of [BLQT21b].** Recent work of Blanc, Lange, Qiao, and Tan also highlights the power of choices in decision tree learning. However, they operate within a stylized theoretical setting. First, they consider a specific scoring function that is based on a notion of *influence* of features, and crucially, computing these scores requires *query access* to the target function (rather than from random labeled samples as is the case in practice). Furthermore, their results only hold with respect to the uniform distribution. These are strong assumptions that limit the practical relevance of their results. In contrast, a primary focus of this work is to be closely aligned with practice, and in particular, our framework captures and generalizes the standard greedy algorithms used in practice.

**Optimal decision trees.** Motivated in part by the surge of interest in interpretable machine learning and the highly interpretable nature of decision trees, there have been numerous works on learning *optimal* decision trees [BD17, VZ17, VZ19, AAV19, ZMP+20, VNP+20, NIPMS18, Ave20, JM20, NF07, NF10, HRS19, LZH+20, DLH+22]. As mentioned in the introduction, this is an NP-complete problem [HR76]—indeed, it is NP-hard to find even an approximately optimal decision tree [Sie08, AH08, ABF+09]. Due to the fundamental intractability of this problem, even highly optimized versions of algorithms are unlikely to match the scalability of standard greedy algorithms. That said, these works implement a variety of optimizations that allow them to build optimal decision trees for many real world datasets when the dataset and feature sizes are in the hundreds and the desired depth is small ($\leq 5$).

Finally, another related line of work is that of *soft* decision trees [IYA12, TAA+19]. These works use gradient-based methods to learn soft splits at each internal node. We believe that one key advantage of our work over these soft trees is in interpretability. With Top-$k$, since the splits are hard (and not

soft), to understand the classification of a test point, it is sufficient to look at only one root-to-leaf path, as opposed to a weighted combination across many.

# 3 The Top-$k$ algorithm

## 3.1 Background and context: Impurity-based algorithms

Greedy decision tree learning algorithms like ID3, C4.5 and CART are all instantiations of Top-$k$ in Figure 1 with $k = 1$ and an appropriate choice of the feature-scoring function $\mathcal{H}$. Those three algorithms all used *impurity-based heuristics* as their feature-scoring function:

**Definition 2** (Impurity-based heuristic). *An impurity function $\mathcal{G} : [0, 1] \to [0, 1]$ is a function that is concave, symmetric about $0.5$, and satisfies $\mathcal{G}(0) = \mathcal{G}(1) = 0$ and $\mathcal{G}(0.5) = 1$. A feature-scoring function $\mathcal{H}$ is an* impurity-based heuristic, *if there is some impurity function $\mathcal{G}$ for which:*

$$\mathcal{H}(S, i) = \mathcal{G} \left( \mathop{\mathbb{E}}_{\boldsymbol{x}, \boldsymbol{y} \sim S} [\boldsymbol{y}] \right) - \mathop{\Pr}_{\boldsymbol{x}, \boldsymbol{y} \sim S} [\boldsymbol{x}_i = 0] \cdot \mathcal{G} \left( \mathop{\mathbb{E}}_{\boldsymbol{x}, \boldsymbol{y} \sim S} [\boldsymbol{y} \mid \boldsymbol{x}_i = 0] \right)$$

$$- \mathop{\Pr}_{\boldsymbol{x}, \boldsymbol{y} \sim S} [\boldsymbol{x}_i = 1] \cdot \mathcal{G} \left( \mathop{\mathbb{E}}_{\boldsymbol{x}, \boldsymbol{y} \sim S} [\boldsymbol{y} \mid \boldsymbol{x}_i = 1] \right)$$

*where in each of the above, $(\boldsymbol{x}, \boldsymbol{y})$ are a uniformly random point from within $S$.*

Common examples for the impurity function include the binary entropy function $\mathcal{G}(p) = -p \log_2(p) - (1 - p) \log_2(1 - p)$ (used by ID3 and C4.5), the Gini index $\mathcal{G}(p) = 4p(1 - p)$ (used by CART), and the function $\mathcal{G}(p) = 2\sqrt{p(1 - p)}$ (proposed and analyzed in [KM99]). We refer the reader to [KM99] for a theoretical comparison, and [DKM96] for an experimental comparison, of these impurity-based heuristics.

Our experiments focus on binary entropy being the impurity measure, but our theoretical results apply to Top-$k$ instantiated with *any* impurity-based heuristic.

## 3.2 Basic theoretical properties of the Top-$k$ algorithm

**Running time.** The key behavioral aspect in which Top-$k$ differs from greedy algorithms is that it is less greedy when trying to determine which coordinate to query. This naturally increases the running time of Top-$k$, but that increase is fairly mild. More concretely, suppose Top-$k$ is run on a dataset $S$ with $n$ points. We can then easily derive the following bound on the running time of Top-$k$, where $\mathcal{H}(S, i)$ is assumed to take $O(n)$ time to evaluate (as it does for all impurity-based heuristics).

**Claim 3.1.** *The running time of Top-$k(\mathcal{H}, S, h)$ is $O((2k)^h \cdot nd)$.*

*Proof.* Let $T_h$ be the number of recursive calls made by Top-$k(\mathcal{H}, S, h)$. Then, we have the simple recurrence relation $T_h = 2kT_{h-1}$, where $T_0 = 1$. Solving this recurrence gives $T_h = (2k)^h$. Each recursive call takes $O(nd)$ time, where the bottleneck is scoring each of the $d$ features. $\square$

We note that any decision tree algorithm, including fast greedy algorithms such as ID3, C4.5, and CART, has runtime that scales exponentially with the depth $h$. The size of a depth-$h$ tree can be $2^h$, and this is of course a lower bound on the runtime as the algorithm needs to output such a tree. In contrast with greedy algorithms (for which $k = 1$), Top-$k$ incurs an additional $k^h$ cost in running time. As mentioned earlier, in practice, we are primarily concerned with fitting small decision trees (e.g., $h = 5$) to the data, as this allows for explainable predictions. In this setting, the additional $k^h$ cost (for small constant $k$) is inexpensive, as confirmed by our experiments.

**The search space of Top-$k$:** We state and prove a simple claim that Top-$k$ returns the *best* tree within its search space.

**Definition 3** (Search space of Top-$k$). *Given a sample $S$ and integers $h, k$, we use $\mathcal{T}_{k,h,S}$ to refer to all trees in the search space of Top-$k$. Specifically, if $h = 0$, this contains all trees with a height of zero (the constant $0$ and constant $1$ trees). For $h \geq 1$, and $\mathcal{I} \subseteq [d]$ being the $k$ coordinates with maximal score, this contains all trees with a root of $x_i$, left subtree in $\mathcal{T}_{k,h-1,S_{x_i=0}}$ and right subtree in $\mathcal{T}_{k,h-1,S_{x_i=1}}$ for some $i \in \mathcal{I}$.*

**Lemma 3.2** (Top-$k$ chooses the most accurate tree in its search space). *For any sample $S$ and integers $h, k$, let $T$ be the output of Top-$k$ with a depth budget of $h$ on $S$. Then*

$$\Pr_{\boldsymbol{x},\boldsymbol{y} \sim S}[T(\boldsymbol{x}) = \boldsymbol{y}] = \max_{T' \in \mathcal{T}_{k,h,S}} \left( \Pr_{\boldsymbol{x},\boldsymbol{y} \sim S}[T'(\boldsymbol{x}) = \boldsymbol{y}] \right).$$

We refer the reader to Appendix A for the proof of this lemma.

# 4 Theoretical bounds on the power of choices

We refer the reader to the Appendix B for most of the setup and notation. For now, we briefly mention a small amount of notation relevant to this section: we use **bold font** (e.g. $\boldsymbol{x}$) to denote random variables. We also use bold font to indicate *stochastic functions* which output a random variable. For example,

$$\boldsymbol{f}(x) := \begin{cases} x & \text{with probability } \frac{1}{2} \\ -x & \text{with probability } \frac{1}{2} \end{cases}$$

is the stochastic function that returns either the identity or its negation with equal probability. To define the data distributions of Theorems 2 and 3, we will give a distribution over the domain, $X$ and the stochastic function that provides the label given an element of the domain.

**Intuition for proof of greediness hierarchy theorem**    To construct a distribution which Top-$k$ fits poorly and Top-$(k + 1)$ fits well, we will partition features into two groups: one group consisting of features with medium correlation to the labels and another group consisting of features with high correlation when taken all together but low correlation otherwise. Since the correlation of features in the former group is larger than that of the latter group unless all features from the latter group are considered, both algorithms will prioritize features from the former group. However, if the groups are sized correctly, then Top-$(k + 1)$ will consider splitting on all features from the latter group, whereas Top-$k$ will not. As a result, Top-$(k + 1)$ will output a decision tree with higher accuracy.

## 4.1 Proof of Theorem 2

For each depth budget $h$ and search branching factor $K$, we will define a hard distribution $\mathcal{D}_{h,K}$ that is learnable to high accuracy by Top-$K$ with a depth of $h$, but not by Top-$k$ with a depth of $h'$ for any $h' < h + K - k$. This distribution will be over $\{0,1\}^d \times \{0,1\}$, where $d = h + K - 1$. The marginal distribution over $\{0,1\}^d$ is uniform, and the distribution over $\{0,1\}$ conditioned on a setting of the $d$ features is given by the stochastic function $\boldsymbol{f}_{h,K}(x)$. All of the results of this section (Theorems 2 and 3) hold when the feature scoring function is *any* impurity-based heuristic.

**Description of $\boldsymbol{f}_{h,K}(x)$.**    Partition $x$ into two sets of variables, $x^{(1)}$ of size $h$ and $x^{(2)}$ of size $K - 1$. Let $\boldsymbol{f}_{h,K}(x)$ be the randomized function defined as follows:

$$\boldsymbol{f}_{h,K}(x) = \begin{cases} \mathrm{Par}_h(x^{(1)}) & \text{with probability } 1 - \varepsilon \\ x_i^{(2)} \sim \mathrm{Unif}[x^{(2)}] & \text{with probability } \varepsilon, \end{cases}$$

where $\mathrm{Unif}[x^{(2)}]$ denotes the uniform distribution on $x^{(2)}$. $\mathrm{Par}_h(x^{(1)})$ is the parity function, whose formal definition can be found in Appendix B.

The proof of Theorem 2 is divided into two parts. First, we prove that when the data distribution is $\mathcal{D}_{h,K}$, Top-$K$ succeeds in building a high accuracy tree with a depth budget of $h$. Then, we show that Top-$k$ fails and builds a tree with low accuracy, even given a depth budget of $h + (K - k - 1)$.

**Lemma 4.1** (Top-$K$ succeeds). *The accuracy of Top-$K$ with a depth of $h$ on $\mathcal{D}_{h,K}$ is at least $1 - \varepsilon$.*

**Lemma 4.2** (Top-$k$ fails). *The accuracy of Top-$k$ with a depth of $h'$ on $\mathcal{D}_{h,K}$ is at most $(1/2 + \varepsilon)$ for any $h' < h + K - k$.*

Proofs of both these lemmas are deferred to Appendix B. Theorem 2 then follows directly from these two lemmas.

## 4.2 Proof of Theorem 3

In this section, we overview the proof Theorem 3. Some of the proofs are deferred to Appendix B.2.

Before proving Theorem 3, we formalize the concept of monotonicity. For simplicity, we assume the domain is the Boolean cube, $\{0,1\}^d$, and use the partial ordering $x \preceq x'$ iff $x_i \leq x'_i$ for each $i \in [d]$; however, the below definition easily extends to the domain being any partially ordered set.

**Definition 4** (Monotone). *A stochastic function, $\boldsymbol{f} : \{0,1\}^d \to \{0,1\}$, is monotone if, for any $x, x' \in \{0,1\}^d$ where $x \preceq x'$, $\mathbb{E}[\boldsymbol{f}(x)] \leq \mathbb{E}[\boldsymbol{f}(x')]$. A data distribution, $\mathcal{D}$ over $\{0,1\}^d \times \{0,1\}$ is said to be monotone if the corresponding stochastic function, $\boldsymbol{f}(x)$ returning $(\boldsymbol{y} \mid \boldsymbol{x} = x)$ where $(\boldsymbol{x}, \boldsymbol{y}) \sim \mathcal{D}$, is monotone.*

To construct the data distribution of Theorem 3, we will combine monotone functions, Majority and Tribes, commonly used in the analysis of Boolean functions due to their extremal properties. See Appendix B.2 for their definitions and useful properties. Let $d = h + K - 1$, and the distribution over the domain be uniform over $\{0,1\}^d$. Given some $x \in \{0,1\}^d$, we use $x^{(1)}$ to refer to the first $h$ coordinates of $x$ and $x^{(2)}$ the other $K - 1$ coordinates. This data distribution is labeled by the stochastic function $\boldsymbol{f}$ given below.

$$\boldsymbol{f}(x) := \begin{cases} \text{Tribes}_h(x^{(1)}) & \text{with probability } 1 - \varepsilon \\ \text{Maj}_{K-1}(x^{(2)}) & \text{with probability } \varepsilon. \end{cases}$$

Clearly $\boldsymbol{f}$ is monotone as it is the mixture of two monotone functions. Throughout this subsection, we'll use $\mathcal{D}_{h,K}$ to refer to the data distribution over $\{0,1\}^d \times \{0,1\}$ where to sample $(\boldsymbol{x}, \boldsymbol{y}) \sim \mathcal{D}$, we first draw $\boldsymbol{x} \sim \{0,1\}^d$ uniformly and then $\boldsymbol{y}$ from $\boldsymbol{f}(\boldsymbol{x})$. The proof of Theorem 3 is a direct consequence of the following two Lemmas, both of which we prove in Appendix B.2.

**Lemma 4.3** (Top-$K$ succeeds). *On the data distribution $\mathcal{D}_{h,K}$, Top-$K$ with a depth budget of $h$ achieves at least $1 - \varepsilon$ accuracy.*

**Lemma 4.4** (Top-$k$ fails). *On the data distribution $\mathcal{D}_{h,K}$, Top-$k$ with a depth budget of $h$ achieves at most $\frac{1}{2} + \varepsilon$ accuracy.*

## 5 Experiments

**Setup for experiments.** At all places, the Top-1 tree that we compare to is that given by `scikit-learn` [PVG$^+$11], which according to their documentation[2], is an optimized version of CART. We run experiments on a variety of datasets from the UCI Machine Learning Repository [DG17] (numerical as well as categorical features) having a size in the thousands and having $\approx 50 - 300$ features after binarization. There were $\approx 100$ datasets meeting these criteria, and we took a random subset of 20 such datasets. We binarize all the datasets – for categorical datasets, we convert every categorical feature that can take on (say) $\ell$ values into $\ell$ binary features. For numerical datasets, we sort and compute thresholds for each numerical attribute, so that the total number of binary features is $\approx 100$. A detailed description of the datasets is given in Appendix C.

We build decision trees corresponding to binary entropy as the impurity measure $\mathcal{H}$. In order to leverage existing engineering optimizations from state-of-the-art optimal decision tree implementations, we implement the Top-$k$ algorithm given in Figure 1 via simple modifications to the PyDL8.5 [ANS20, ANS21] codebase[3]. Details about this are provided in Appendix D. Our implementation of the Top-$k$ algorithm and other technical details for the experiments are available at `https://github.com/SullivanC19/pydl8.5-topk`.

### 5.1 Key experimental findings

**Small increments of $k$ yield significant accuracy gains.** Since the search space of Top-$k$ is a superset of that of Top-1 for any $k > 1$, the training accuracy of Top-$k$ is guaranteed to be larger. The primary objective in this experiment is to show that Top-$k$ can outperform Top-1 in terms of test accuracy as well. Figure 2 shows the results for Top-1 versus Top-$k$ for $k = 2, 3, 4, 8, 12, 16, d$. Each

---

[2]https://scikit-learn.org/stable/modules/tree.html#tree-algorithms-id3-c4-5-c5-0-and-cart
[3]https://github.com/aia-uclouvain/pydl8.5

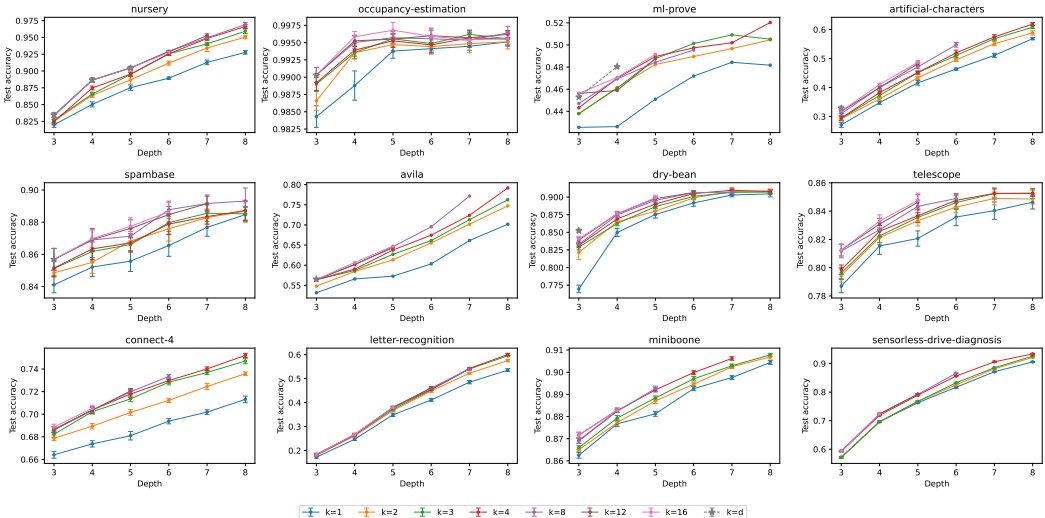

Figure 2: Test accuracy comparison between Top-$k$ for various values of $k$. We can see that Top-$(k+1)$ generally obtains higher accuracy than Top-$k$, and in some cases (e.g., nursery), Top-8/16's accuracy is even comparable to the optimal tree (Top-$d$). Missing points in the plots correspond to settings that did not terminate within a sufficiently large time limit. All plots are averaged over 10 random train-test splits (except avila and ml-prove that have pre-specified splits) with confidence intervals plotted for 2 standard deviations.

plot is a different dataset, where on the x-axis, we plot the depth of the learned decision tree, and on the y-axis, we plot the test accuracy. Note that $k = d$ corresponds to the DL8.5 optimal decision tree. We can clearly observe that the test accuracy increases as $k$ increases—in some cases, the gain is $> 5\%$ (absolute). Furthermore, for (smaller) datasets like nursery, for which we were able to run $k = d$, the accuracy of Top-8/16 is already very close to that of the optimal tree.

Lastly, since Top-$k$ invests more computation towards fitting a better tree on the training set, its training time is naturally longer than Top-1. However, Figure 6 in Appendix E, which plots the training time, shows that the slowdown is mild.

**Top-$k$ scales much better than optimal decision tree algorithms.** Optimal decision tree algorithms suffer from poor runtime scaling. We empirically demonstrate that, in comparison, Top-$k$ has a significantly better scaling in training time. Our experiments are identical to those in Figures 14 and 15 in the GOSDT paper [LZH+20], where two notions of scalability are considered. In the first experiment, we fix the number of samples and gradually increase the number of features to train the decision tree. In the second experiment, we include all the features, but gradually increase the number of training samples. The dataset we use is the FICO [FGI+18] dataset, which has a total of 1000 samples with 1407 binary features. We plot the training time (in seconds) versus number of features/samples for optimal decision tree algorithms (MurTree, GOSDT) and Top-$k$ in Figure 3. We do this for depth $= 4, 5, 6$ (for GOSDT, the regularization coefficient $\lambda$ is set to $2^{-\text{depth}}$). We observe that the training time for both MurTree and GOSDT increases dramatically compared to Top-$k$, in both experiments. In particular, for depth $= 5$, both MurTree and GOSDT were unable to build a tree on 300 features within the time limit of 10 minutes, while Top-16 completed execution even with all 1407 features. Similarly, in the latter experiment, GOSDT/MurTree were unable to build a depth-5 tree on 150 samples within the time limit, while Top-16 comfortably finished execution even on 1000 samples. These experiments demonstrates the scalability issues with optimal tree algorithms. Coupled with the accuracy gains seen in the previous experiment, Top-$k$ can thus be seen as achieving a more favorable tradeoff between training time and accuracy.

We note, however, that various optimization have been proposed to allow optimal decision tree algorithms to scale to larger datasets. For example, a more recent version of GOSDT has integrated a guessing strategy using reference ensembles which guides the binning of continuous features, tree

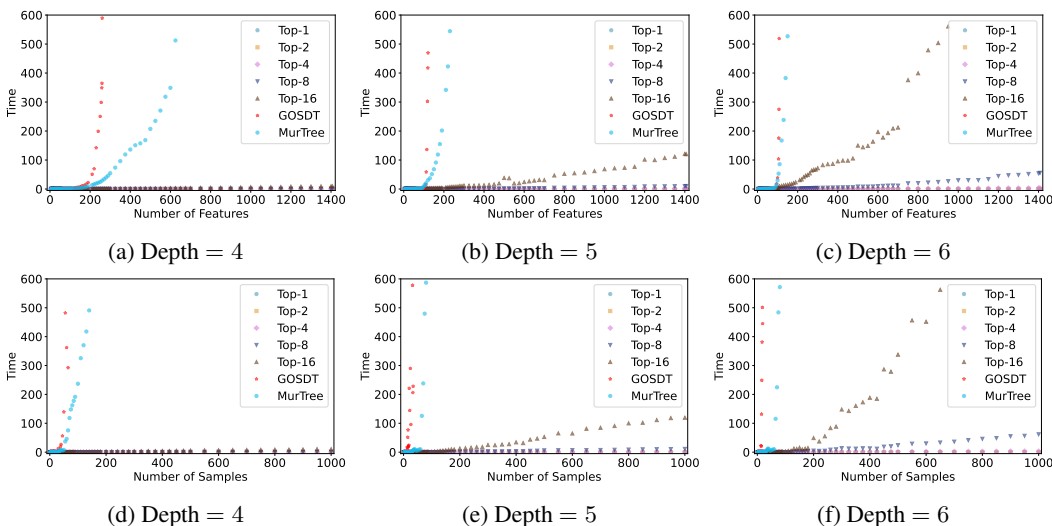

Figure 3: Training time comparison between Top-$k$ and optimal tree algorithms. As the number of features/samples increases, both GOSDT and MurTree scale poorly compared to Top-$k$, and beyond a threshold, do not complete execution within the time limit.

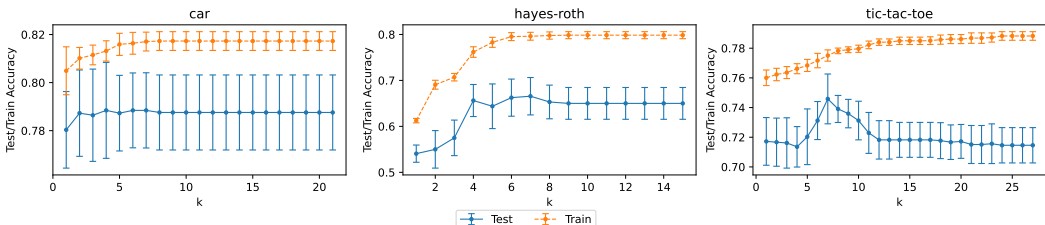

Figure 4: Test accuracy plateaus for large $k$. All runs averaged over 10 random train-test splits with maximum depth fixed to 3.

size, and search [MZA$^+$]. Many of these optimizations are generally applicable across optimal tree algorithms and could be combined with Top-$k$ for further improvement in performance.

**Increasing $k$ beyond a point does not improve test accuracy.** In our experiments above, we ran Top-$k$ only till $k = 16$: in Figure 4, we show that increasing $k$ to very large values, which increases runtime, often does not improve test accuracy, and in some cases, may even *hurt* due to overfitting. For 3 datasets – car, hayes-roth and tic-tac-toe – we plot train and test error as a function of $k$. Naturally, the train accuracy monotonically increases with $k$ in each plot. However, for both car and hayes-roth, we can observe that the test accuracy first increases and then plateaus. Interestingly, for tic-tac-toe, the test accuracy first increases and then *decreases* as we increase $k$. These experiments demonstrate that selecting too large of a $k$, as optimal decision tree algorithms do, is a waste of computational resources and can even hurt test accuracy via overfitting.

## 6 Conclusion

We have shown how popular and empirically successful greedy decision tree learning algorithms can be improved with *the power of choices*: our generalization, Top-$k$, considers the $k$ best features as candidate splits instead of just the single best one. As our theoretical and empirical results demonstrate, this simple generalization is powerful and enables significant accuracy gains while preserving the efficiency and scalability of standard greedy algorithms. Indeed, we find it surprising that such a simple generalization has not been considered before.

There is much more to be explored and understood, both theoretically and empirically; we list here a few concrete directions that we find particularly exciting and promising. First, we suspect that power

of choices affords more advantages over greedy algorithms than just accuracy gains. For example, an avenue for future work is to show that the trees grown by Top-$k$ are more *robust to noise.* Second, are there principled approaches to the automatic selection of the greediness parameter $k$? Can the optimal choice be inferred from a few examples or learned over time? This opens up the possibility of new connections to machine-learned advice and algorithms with predictions [MV20], an area that has seen a surge of interest in recent years. Finally, as mentioned in the introduction, standard greedy decision tree algorithms are at the very heart of modern tree-based ensemble methods such as XGBoost and random forests. A natural next step is to combine these algorithms with Top-$k$ and further extend the power of choices to these settings.

## Acknowledgements

We thank the NeurIPS reviewers and AC for their detailed and helpful feedback.

Guy and Li-Yang are supported by NSF awards 1942123, 2211237, 2224246 and a Google Research Scholar award. Jane is supported by NSF Graduate Research Fellowship under Grant No. 2141064, NSF Awards CCF-2006664, DMS-2022448, and Microsoft. Mo is supported by a Stanford Interdisciplinary Graduate Fellowship and a Stanford Data Science Scholarship. Chirag is supported by Moses Charikar and Greg Valiant's Simons Investigator Awards.

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
