# A  Proofs deferred from Section 3

*Proof of Lemma 3.2.* By induction: When $h = 0$, the only trees in the search space are the constant 0 and constant 1 functions. Top-$k$ returns which of these two trees is the most accurate.

When $h \geq 1$, let $T'$ be a tree with maximal accuracy within $\mathcal{T}_{k,h,S}$. As $T'$ is in the search space, its root must be one of the $k$ coordinates with maximal score which form the candidate set $\mathcal{I}$.

For each coordinate $i \in \mathcal{I}$, the candidate tree $T_i$ satisfies

$$\Pr_{\boldsymbol{x},\boldsymbol{y}\sim S}[T_i(\boldsymbol{x}) \neq \boldsymbol{y}] = \Pr_{\boldsymbol{x}\sim S}[x_i = 0]\Pr_{\boldsymbol{x},\boldsymbol{y}\sim S}[T_{i0}(\boldsymbol{x}) \neq \boldsymbol{y}] + \Pr_{\boldsymbol{x}\sim S}[x_i = 1]\Pr_{\boldsymbol{x},\boldsymbol{y}\sim S}[T_{i1}(\boldsymbol{x}) \neq \boldsymbol{y}],$$

where $T_{i0}$ and $T_{i1}$ are the left and right subtrees of $T_i$ respectively. Each of $T_{i0}$ and $T_{i1}$ is an output of Top-$k$ with a depth budget of $h - 1$. We assume as the inductive hypothesis that each of these trees minimizes error among all trees in $\mathcal{T}_{k,h-1,S_{x_i=0}}$ and $\mathcal{T}_{k,h-1,S_{x_i=1}}$ respectively; therefore the candidate $T_i$ minimizes error among all trees in $\mathcal{T}_{k,h,S}$ that have $x_i$ at the root. Since Top-$k$ chooses the most accurate of the $T_i$'s, it follows that the chosen tree minimizes error among all trees in $\mathcal{T}_{k,h,S}$. $\qquad\square$

# B  Proofs deferred from Section 4

**Setup and notation:**  We use $\mathbb{1}[\cdot]$ for the indicator function, and $[d]$ to refer to the set $\{1, \ldots, d\}$.

For brevity, we will make two simplifying assumptions about Top-$k$:

1. We will assume Top-$k$ builds *non-redundant* trees, meaning on every root-to-leaf path, each coordinate is queried at most once. This is easy to enforce in the pseudocode: at each step, the algorithm can track a set $Q$ of the coordinates already queried along this path, and pick the top-$k$ coordinates according to the feature score function among $[d] \setminus Q$. For brevity, we do not include that modification to the pseudocode in Figure 1.

2. We assume that Top-$k$ always build *complete* trees (i.e every root-to-leaf path has depth exactly $h$). This is without loss of generality, as whenever Top-$k$ stops early, it does so because it has already achieved perfect accuracy on that path.

Furthermore, Top-$k$ only uses the information in its sample in two ways: first, it uses the sample to compute the feature scoring function $\mathcal{H}(S, i)$. Second, when $h = 0$, it uses the sample to determine whether the constant 0 or constant 1 fits the sample better. Both of these are "statistical queries" [Kea98], meaning the interaction the algorithm receives from the sample is simply the expectations $\mathbb{E}_{(\boldsymbol{x},\boldsymbol{y})\sim S}[\phi_i(\boldsymbol{x},\boldsymbol{y})]$ where $\phi_1, \ldots, \phi_t : \{0,1\}^{d+1} \to [0,1]$ are a sequence of queries. For any $\varepsilon, \delta > 0$, by a standard concentration argument and union bound, for large enough sample size $n \geq n(\varepsilon, \delta)$,

$$\Pr_{\boldsymbol{S}\sim\mathcal{D}^n}\left[\max_{i\in[t]}\left|\mathbb{E}_{(\boldsymbol{x},\boldsymbol{y})\sim\boldsymbol{S}}[\phi_i(\boldsymbol{x},\boldsymbol{y})] - \mathbb{E}_{(\boldsymbol{x},\boldsymbol{y})\sim\mathcal{D}}[\phi_i(\boldsymbol{x},\boldsymbol{y})]\right| \geq \varepsilon\right] \leq \delta.$$

Therefore, for sufficiently large sample size, we are free to assume that when the algorithm computes $\mathbb{E}_{(\boldsymbol{x},\boldsymbol{y})\sim S}[\phi_i(\boldsymbol{x},\boldsymbol{y})]$, it receives $\mathbb{E}_{(\boldsymbol{x},\boldsymbol{y})\sim\mathcal{D}}[\phi_i(\boldsymbol{x},\boldsymbol{y})]$ with high probability. This is a standard argument (c.f. [KM96]), and so we will work directly with expectations from $\mathcal{D}$ in our proof to ease notation.

Recall that Theorems 1 to 3 hold whenever the feature scoring function is an impurity-based heuristic.As our data distribution is uniform on the input, we are able to use the following fact and simultaneously prove results for all impurity-based heuristic:

**Fact B.1** (Proposition 7.7 of [BLT20b])**.** *If the scoring function is* any *impurity-based heuristic, and the data distribution is uniform over inputs ($\boldsymbol{x}$ is uniform when $(\boldsymbol{x}, \boldsymbol{y}) \sim \mathcal{D}$), then the score of a coordinate $i$ is monotone increasing with its correlation with the label, $\mathbb{E}_{(\boldsymbol{x},\boldsymbol{y})\sim\mathcal{D}}[\boldsymbol{x}_i\boldsymbol{y}]$.*

Intuitively, Fact B.1 means that, when analyzing Top-$k$ on uniform data distributions, we are free to replace the "$k$ coordinates with largest scores" with the "$k$ coordinates with largest correlations."

## B.1  Proofs deferred from Section 4.1

The stochastic function $f_{h,K}$ used throughout Lemma 4.1 and Lemma 4.2 combines a function that outputs a random one of $k$ features with the $h$-wise parity function.

**Definition 5** (Parity). *The* parity *function of $\ell$ variables, indicated by* $\mathrm{Par}_\ell : \{0,1\}^\ell \to \{0,1\}$, *returns*

$$\mathrm{Par}_\ell(x) := \left( \sum_{i \in [\ell]} x_i \right) \mod 2.$$

**Fact B.2** (Computing any function with a complete tree). *Let $f : \{0,1\}^d \to \{0,1\}$ be any function that only depends on the first $h$ variables, meaning there is some $g : \{0,1\}^h \to \{0,1\}$ such that:*

$$f(x) = g(x_{[1:h]})$$

*for all $x \in \{0,1\}^d$. Let $T$ be any non-redundant complete tree of depth-$h$ in which every internal node is one of the first $h$ coordinates. Then, there is a way to label the leaves of $T$ such that $T$ exactly computes $f$.*

*Proof.* Since $T$ is non-redundant, each coordinate is queried at most once on each root-to-leaf path. $T$ is complete and depth-$h$, so each of the first $h$ coordinates must be queried *exactly* once on each root-to-leaf path. Therefore, each leaf of $T$ corresponds to exactly one way to set the first $k$ coordinates of $x$. If the leaf is labeled by the output of $g$ given those first $k$ coordinates, $T$ will exactly compute $f$. □

*Proof of Lemma 4.1.* The function $\mathrm{Par}_h(x^{(1)})$ is a $(1-\varepsilon)$-approximation to $f$, so it suffices to show that the depth-$h$ tree for $\mathrm{Par}_h(x^{(1)})$ is within the search space of Top-$K$ when run to a depth of $h$. Then we can apply Lemma 3.2 to reach the desired result.

There are only $K-1$ variables not in $x^{(1)}$, so each set of $K$ candidate variables must contain some variable in $x^{(1)}$. Since Top-$K$ is non-redundant, this must be a variable that has not yet been queried higher in the tree. Thus, at every step Top-$K$ will always try a candidate variable that reduces the number of relevant $x^{(1)}$-variables by 1. It follows that the complete nonadaptive tree of depth $h$, containing all the variables of $x^{(1)}$, is within the search space, so by Fact B.2 there is a tree in the search space that computes $\mathrm{Par}_h(x^{(1)})$ exactly. Then the accuracy of the output must be at least the total accuracy of this tree, which is $(1-\varepsilon)$. □

*Proof of Lemma 4.2.* Conditioned on any setting of $< k$ variables, for any variable $x_i$ in $x^{(2)}$, $\mathbb{E}[f(x)x_i] \geq 1/k$. Similarly, for any variable $x_j$ in $x^{(1)}$, $\mathbb{E}[f(x)x_j] = 0$. By Fact B.1, at every node the variables of $x^{(2)}$ that have not yet been queried all rank ahead of the variables of $x^{(1)}$. Thus, if at most $K-k$ variables have already been queried, the remaining $k$ most-correlated candidates will all be from $x^{(2)}$, so no variable in $x^{(1)}$ will be considered. Thus, at least $K-k$ variables from $x^{(2)}$ will be placed in every path.

Since the depth budget $h'$ is smaller than $h + K - k$ and at least $K - k$ variables from $x^{(2)}$ are placed in every path, no path can contain all of the $h$ variables of $x^{(1)}$. The value of $\mathrm{Par}_h(x^{(1)})$ is 0 with probability 1/2 and 1 with probability 1/2 conditioned on the values of any set of variables smaller than $h$. Therefore, the tree built by Top-$k$ cannot achieve accuracy better than 1/2 on the parity portion of the function (and thus have accuracy better than $(1/2 + \varepsilon)$ overall).

□

## B.2  Proofs deferred from Section 4.2

The data distribution showing the accuracy separation between Top-$K$ and Top-$k$ is formed by combining the Majority and Tribes functions.

**Definition 6** (Majority). *The* majority *function of $\ell$ variables, indicated by* $\mathrm{Maj}_\ell : \{0,1\}^\ell \to \{0,1\}$, *returns*

$$\mathrm{Maj}_\ell(x) := \mathbb{1}[\text{at least half of $x$'s coordinates are } 1].$$

**Definition 7** (Tribes). *For any input length $\ell$, let $w$ be the largest integer such that $(1 - 2^{-w})^{\ell/w} \leq 1/2$. For $x \in \{0,1\}^\ell$, let $x^{(1)}$ be the first $w$ coordinates, $x^{(2)}$, the second $w$, and so on. $\mathrm{Tribes}_\ell$ is defined as*

$$\mathrm{Tribes}_\ell(x) := (x_1^{(1)} \wedge \cdots \wedge x_w^{(1)}) \vee \cdots \vee (x_1^{(t)} \wedge \cdots \wedge x_w^{(t)}) \qquad \text{where } t := \left\lfloor \frac{\ell}{w} \right\rfloor.$$

For our purposes, it is sufficient to know a few simple properties about Tribes. These are all proven in [O'D14, §4.2].

**Fact B.3** (Properties of Tribes).

1. *$\mathrm{Tribes}_\ell$ is monotone.*

2. *$\mathrm{Tribes}_\ell$ is nearly balanced:*

$$\mathop{\mathbb{E}}_{\boldsymbol{x} \sim \{0,1\}^\ell}[\mathrm{Tribes}_\ell(\boldsymbol{x})] = \frac{1}{2} \pm o(1)$$

    *where the $o(1)$ term goes to 0 as $\ell$ goes to $\infty$.*

3. *All variables in $\mathrm{Tribes}_\ell$ have small correlation: For each $i \in [\ell]$,*

$$\mathrm{Cov}_{\boldsymbol{x} \sim \{0,1\}^\ell}[\boldsymbol{x}_i, \mathrm{Tribes}_\ell(\boldsymbol{x})] = O\left(\frac{\log \ell}{\ell}\right).$$

Indeed, the famous KKL inequality implies that any function with the first and second property has a variable with correlation at least $\Omega(\log \ell / \ell)$ [KKL88]. Our construction uses Tribes exactly because it has the minimum correlations among functions with the above properties (up to constants). In contrast, we use Majority because its correlations are as *large* as possible, which will "trick" Top-$k$ into building a bad tree.

With the above definitions in-hand, we are able to provide proofs of the following two lemmas:

*Proof of Lemma 4.3.* This proof is very similar to that of Lemma 4.1: Once again, we observe the tree computing $(x \mapsto \mathrm{Tribes}_h(x^{(1)}))$ has at least $1 - \varepsilon$ accuracy with respect to $\mathcal{D}_{h,K}$. By Lemma 3.2, it is sufficient to prove such a tree is in the search space.

By Fact B.2, any non-redundant complete tree of depth $h$ that only queries the first $h$ coordinates of its input will compute the function $(x \mapsto \mathrm{Tribes}_h(x^{(1)}))$ whenever the leaves are appropriately labeled. Therefore, we only need to prove such a tree is in the search space $\mathcal{T}_{K,h,\mathcal{D}}$. There are only $K - 1$ coordinates that are *not* one of the first $h$ corresponding to $x^{(1)}$. Therefore, within any non-redundant set of $K$ coordinates, at least one must be a non-redundant coordinate from the first $h$. This implies one of the desired trees is in the search space. $\qquad \square$

*Proof of Lemma 4.4.* Let $T$ be the tree returned by Top-$k$. Consider any root-to-leaf path of $T$ that does *not* query any of the first $h$ coordinates (those within $x^{(1)}$). Recall that, with probability $(1 - \varepsilon)$, the label is given by $\mathrm{Tribes}_h(x^{(1)})$. On this path, the label of $T$ does not depend on any of the coordinates within $x^{(1)}$. Therefore,

$$\Pr_{(\boldsymbol{x},\boldsymbol{y}) \sim \mathcal{D}_{h,K}}[T(\boldsymbol{x}) = \boldsymbol{y} \mid \boldsymbol{x} \text{ follows this path}]$$

$$= (1 - \varepsilon) \cdot \Pr_{(\boldsymbol{x},\boldsymbol{y}) \sim \mathcal{D}_{h,K}}[T(\boldsymbol{x}) = \mathrm{Tribes}_h(\boldsymbol{x}^{(1)}) \mid \boldsymbol{x} \text{ follows this path}]$$

$$+ \varepsilon \cdot \Pr_{(\boldsymbol{x},\boldsymbol{y}) \sim \mathcal{D}_{h,K}}[T(\boldsymbol{x}) = \mathrm{Maj}_K(\boldsymbol{x}^{(2)}) \mid \boldsymbol{x} \text{ follows this path}]$$

$$\leq (1 - \varepsilon) \cdot \left(\frac{1}{2} + o(1)\right) + \varepsilon \cdot 1 \leq \frac{1 + \varepsilon}{2} + o(1)$$

where the last line follows because $\mathrm{Tribes}_h$ is nearly balanced (Fact B.3). As the distribution over $\boldsymbol{x}$ is uniform, each leaf is equally likely. Therefore, if only $p$-fraction of root-to-leaf paths of $T$ query at least one of the first $h$ coordinates, then,

$$\Pr_{(\boldsymbol{x},\boldsymbol{y}) \sim \mathcal{D}_{h,K}}[T(\boldsymbol{x}) = \boldsymbol{y}] \leq (1 - p) \cdot \left(\frac{1 + \varepsilon}{2} + o(1)\right) + p \cdot 1 \leq \frac{1}{2} + \frac{p}{2} + \frac{\varepsilon}{2} + o(1)$$

Our goal is to prove the tree returned by Top-$k$ achieves at most $\frac{1}{2} + \varepsilon$ accuracy. Therefore, it is enough to prove that $p = o(1)$. Indeed, we will prove that $p \leq O(K^{-2})$.

Here, we apply [BLT20b, Lemma 7.4], which was used to show that Top-1 fails to build a high accuracy tree. They used a different data distribution, but that particular Lemma still applies to our setting. They prove that a random root-to-leaf path of $T$ satisfies the following with probability at least $1 - O(K^{-2})$: If the length of this path is less than $O(K/\log K)$, at any point along that path, all coordinates within $x^{(2)}$ that have not already been queried have correlation at least $\frac{1}{100\sqrt{k}}$.

That Lemma will be useful for proving Top-$k$ fails with the following parameter choices.

1. By setting $K \geq \Omega(h \log h)$, we can ensure all root-to-leaf paths in $T$ have length at most $O(K/\log K)$, so [BLT20b, Lemma 7.4] applies.

2. By setting $K \leq O(h^2/(\log h)^2)$, we can ensure that all the coordinates within $x^{(1)}$ have correlation less than $\frac{1}{100\sqrt{k}}$ (Fact B.3). This means that all non-redundant coordinates within $x^{(2)}$ have more correlation than those within $x^{(1)}$.

3. By setting $k \leq K - h$, we ensure at all nodes along every path, there are at least $k$ coordinates within the last $K - 1$ coordinates (those corresponding to $x^{(2)}$), that have not already been queried. With probability at least $1 - O(K^{-2})$ over a random path, those all have more correlation than all coordinates within $x^{(1)}$, so Top-$k$ won't query any of the $h$ coordinates within $x^{(1)}$.

We conclude that, with probability at least $1 - O(K^{-2})$ over a random path in $T$, that path does not query any of the first $h$ variables. As a result, the accuracy of $T$ is at most $\frac{1+\varepsilon}{2} + o(1) \leq \frac{1}{2} + \varepsilon$. $\quad\square$

## C   Details about datasets used in Section 5

| Name | Type | Size (#train/#test) | #feats | #binary feats | #classes |
|---|---|---|---|---|---|
| connect-4 | C | 67557 (54045/13512) | 42 | 126 | 3 |
| nursery | C | 12960 (10368/2592) | 8 | 27 | 5 |
| letter-recognition | C | 19999 (15999/4000) | 16 | 256 | 26 |
| car | C | 1728 (1382/346) | 6 | 21 | 4 |
| kr-vs-kp | C | 3196 (2556/640) | 36 | 73 | 2 |
| hiv-1-protease | C | 6590 (5272/1318) | 8 | 160 | 2 |
| molecular-biology-splice | C | 3190 (2552/638) | 60 | 287 | 3 |
| monks-1 | C | 556 (444/112) | 6 | 17 | 2 |
| hayes-roth | C | 160 (128/32) | 4 | 15 | 3 |
| tic-tac-toe | C | 958 (766/192) | 9 | 27 | 2 |
| artificial-characters | N | 10218 (8174/2044) | 7 | 91 | 10 |
| telescope | N | 19020 (15216/3804) | 10 | 100 | 2 |
| spambase | N | 4601 (3680/921) | 57 | 57 | 2 |
| dry-bean | N | 13611 (10888/2723) | 16 | 96 | 7 |
| occupancy-estimation | N | 10129 (8103/2026) | 16 | 86 | 4 |
| miniboone | N | 130064 (104051/26013) | 50 | 100 | 2 |
| sensorless-drive-diagnosis | N | 58509 (46807/11702) | 48 | 96 | 11 |
| ml-prove | N | 6118 (4588/1530) | 51 | 51 | 6 |
| avila | N | 20867 (10430/10437) | 10 | 100 | 12 |
| taiwanese-bankruptcy | N | 6819 (5455/1364) | 95 | 95 | 2 |
| credit-card | N | 30000 (24000/6000) | 23 | 88 | 2 |
| electrical-grid-stability | N | 10000 (8000/2000) | 13 | 91 | 2 |
| FICO | N | 1000 (900/100) | 23 | 1407 | 2 |

Table 1: Dataset characteristics. In the Type column, C stands for Categorial and N stands for Numerical.

Table 1 provides complete details regarding all the datasets we used in our experiments. For datasets that do not provide an explicit train/test split, we randomly compute ten 80:20 splits, and average our results over these splits. The column #feats has the number of raw attributes in each dataset, while the column #binary feats has the number of features we obtain after converting these raw attributes to binary-valued attributes. For categorical datasets, we encode a categorical attribute taking on $l$ distinct values to $l$ binary attributes. For numerical datasets, we sort and compute thresholds for each numerical attribute. The number of thresholds is so selected that the total number of binary attributes does not exceed 100.

## D    Implementation details for the Top-$k$ algorithm

Our implementation of Top-$k$ makes use of the DL8.5 algorithm implementation from [ANS21]. DL8.5 is an optimal classification tree search algorithm which utilizes caching and branch-and-bound optimization to avoid repeated computation and prune large sections of the search space that would yield suboptimal trees [ANS20], similar to MurTree [DLH+22]. To get our optimized Top-$k$ algorithm, we modify DL8.5 to only consider the first $k$ feature splits of each recursive state in descending order of information gain and with ties broken by feature index.

There were two other optimizations made by the DL8.5 algorithm implementation that would have led to different results. These optimizations are (1) fast computation of depth-two optimal trees and (2) similarity-based lower bounding. These optimizations were disabled.

## E    Training time comparison

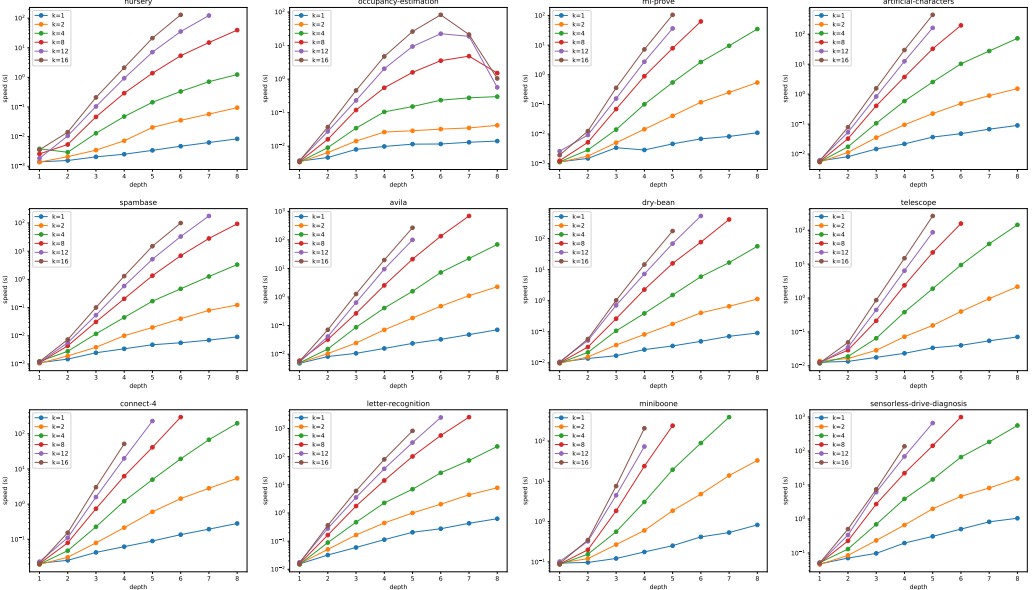

Figure 6: Training time comparison between Top-1 and Top-$k$. We can see that the blowup in training time when compared to Top-1 is relatively mild. In particular, for $k = 2$, we are able to go all the way up until depth-8 trees within 1 second in almost all cases. Even $k = 4, 8$ finishes execution for depth-5 trees within $\approx 20$ seconds for majority of the datasets. Interestingly, in the case of occupancy-estimation, we can see that the training times get *faster* at the larger depths. This is an artefact of the optimized branch-and-bound implementation of DL8.5, which stops branching once it discovers a subtree with no errors. We expect these perfect subtrees to become more prevalent when considering higher depth trees and when there are fewer points to be classified.

Opt-Top-$k(\mathcal{H}, S, h, ub)$:

**Given:** A feature scoring function $\mathcal{H}$, a labeled sample set $S$ over $d$ dimensions, depth budget $h$, and upper bound on misclassification error $ub$.

**Output:** Decision tree of depth $h$ that approximately fits $S$.

1. If $h = 0$, or if every point in $S$ has the same label, return the constant function with the best accuracy w.r.t. $S$.

2. **If $(S, d)$ is in the cache:**
   (a) **Let $T_c$ and $ub_c$ be the cached tree and upper bound.**
   (b) **If $T_c \neq$ NO-TREE then return $T_c$.**
   (c) **If $T_c =$ NO-TREE and $ub \leq ub_c$ then return NO-TREE.**

3. **Let $T^*$ be NO-TREE.**

4. **Let $b^*$ be $ub + 1$.**

5. Let $\mathcal{I} \subseteq [d]$ be the set of $k$ coordinates maximizing $\mathcal{H}(S, i)$.

6. For each $i \in \mathcal{I}$:
   (a) Let $T_i$ be the tree with

   $$\text{Root} = x_i$$
   $$\text{Left subtree} = \text{Opt-Top-}k(\mathcal{H}, S_{x_i=0}, h - 1, b^* - 1)$$

   (b) **If the left subtree is NO-TREE then continue.**
   (c) **Let $b_L$ be the misclassification error of the left subtree w.r.t. $S_{x_i=0}$.**
   (d) **If $b_L \leq b^*$** we define the right subtree of $T_i$

   $$\text{Right subtree} = \text{Opt-Top-}k(\mathcal{H}, S_{x_i=1}, h - 1, b^* - 1 - b_L)$$

   (e) **If the right subtree is NO-TREE then continue.**
   (f) **Let $b_R$ be the misclassification error of the right subtree w.r.t. $S_{x_i=1}$.**
   (g) **If $b_L + b_R < b^*$:**
       i. **Let $T^* = T_i$.**
       ii. **Let $b^* = b_L + b_R$.**
   (h) **If $b_L + b_R = 0$ then break.**

7. **Add $(S, d)$ to the cache with value $(T^*, ub)$.**

8. Return $T^*$.

Figure 5: The optimized Top-$k$ algorithm is equivalent to the Top-$k$ algorithm described in Figure 1 but with caching and pruning optimizations that make it significantly faster in practice. These changes are bolded and highlighted in blue.

# F   Accuracy comparison with Top-1 – further plots

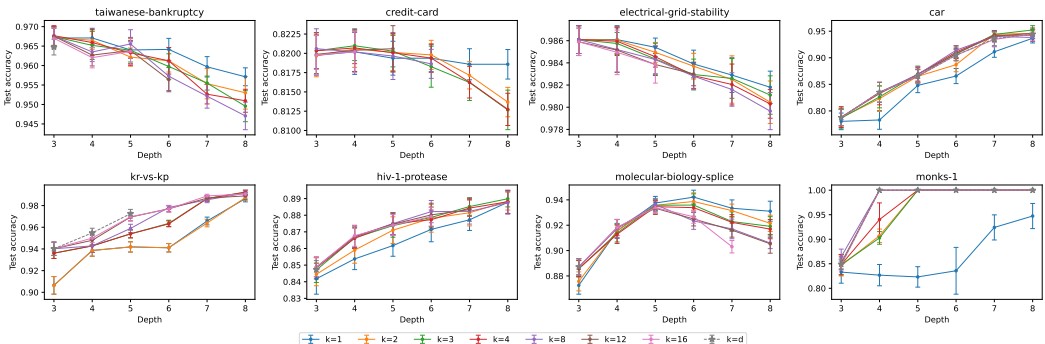

Figure 7: Test accuracy comparison between Top-1 and Top-$k$.

We provide plots from our experiments on a further few datasets comparing the test accuracy of Top-$k$ and Top-1 in Figure 7. In the case of taiwanese-bankruptcy, credit-card and electrical-grid-stability, we can observe that Top-1 is outperforming Top-$k$. However, we believe that this is because the learning problem in this regime is extremely susceptible to overfitting. In particular, we can see that Top-1 is itself not consistently improving with increasing depth. Concretely, increasing depth beyond 3 is already causing Top-1 to overfit, and hence we would expect Top-$k$ to suffer from overfitting even more. In the case of the remaining datasets (which all happen to be categorical), while the numbers might not be monotonically getting better with increasing $k$, we can still observe that there is always some value of $k > 1$ which is outperforming $k = 1$ (except for molecular-biology-splice, for which this is still the case till depth 6). This lends further support to our proposition of incorporating $k$ as an additional hyperparameter to tune while training decision trees greedily.