# OpenReview forum: "Harnessing the power of choices in decision tree learning"
_NeurIPS.cc/2023/Conference — NeurIPS 2023 poster_

### Official Review · Reviewer_8rCb · 2023-06-20

**Soundness:** 3 good
**Presentation:** 3 good
**Contribution:** 2 fair
**Rating:** 6
**Confidence:** 3

**Summary:**

The paper is about a new method for machine learning using decision trees. Namely, the authors propose an algorithm that can span the spectrum of the methods that are currently used:
(i) traditional decision tree learning algorithms that choose features and splitting points based on a criterion that is optimized greedily (e.g., gini impurity), and
(ii) optimal decision tree learning, which performs an exhaustive search for the best possible short decision tree (we should also have fairly few candidate features for splitting).

In particular, the authors allow a parameter k that controls the amount of splits that will be considered at every node. In other words, the authors consider the top k greedy choices at every node. This allows one to consider exponentially many trees, and thus choose the best tree among those considered. Note that when k=1, the algorithm reduces to the standard greedy algorithms such as C4.5 and CART, whereas when k=d (where d is the dimension of the instances), we obtain an algorithm that constructs an optimal decision tree.

The authors have some nice results to complement the proposed approach. For example, in Theorem 1 of the paper, it is shown that there are data distributions where a (k+1) branching factor would allow accuracy of 1-ε, while a branching factor of k can achieve accuracy at most 1/2 + ε. The authors also show (in Theorem 2) a generalization of the previously mentioned theorem. Eventually the authors show experimentally that the proposed algorithm behaves well in an experimental setting with the solutions provided between the two extremes of optimal decision tree learning and pure greedy methods that have been used for decades. In other words, through the experiments, the authors highlight the tradeoff one would be willing to explore by spending some extra time on constructing decision trees that can potentially allow the user to achieve better accuracy, while at the same time one would be able to create much deeper trees, much faster, compared to optimal decision trees.

After the rebuttal: Upgraded the score from borderline accept to weak accept since I was satisfied with the responses that I received and largely with the responses that the authors gave to the comments and issues that were raised by the other reviewers.

**Strengths:**

I think this is a well-written paper that is also backed up with theoretical analysis and is also validated (to a large extent) in the experiments. It is an interesting algorithm that appears to be improving the accuracy of the learned decision trees in many practical datasets, while at the same time the increase in performance is probably tolerable. The paper also has a fairly nice summary of related work.

**Weaknesses:**

The paper is not based on any striking new idea. While the novelty is limited, it is nevertheless interesting.

**Questions:**

Q1. How many runs did you use for the results in Figure 2? Are the conclusions based on statistically significant observations?

Q2. On a related note, in Figure 2, in the case of spambase (and some other cases), MurTree that calculates the optimal decision tree, provides a solution with worse accuracy compared to what the authors achieve with their method. Can the authors shed some light in this? Also, another observation has to do with the fact that on the same graph (spambase), at depth 8, top-2 outperforms top-3. Is there an explanation for this?

**Limitations:**

I think the paper is ok.

---

> ### Author Rebuttal · Authors · 2023-08-09
>
> Reviewer 8rCb Comment 1: The paper is not based on any striking new idea. While the novelty is limited, it is nevertheless interesting.
>
> Response to Reviewer 8rCb Comment 1: We believe the simplicity of our ideas, combined with theoretical performance guarantees and the fact that these ideas work well empirically, are strengths of our work. Regarding novelty, prior work has focused only on either end of the spectrum: greedy tree construction or optimal tree construction. In our framework, greedy tree construction corresponds to $k=1$ and optimal tree construction corresponds to $k=d$. Our theoretical results show that there always exists a dataset for which increasing $k$ can be valuable. We believe that our work unifies the approaches at either end of the spectrum and provides useful theoretical results and examples that can guide the choice of $k$ to interpolate between fast, greedy tree construction and slow, optimal tree construction. We consider our theoretical contribution nontrivial and believe that it strongly motivates the Top-$K$ algorithm. As such, we believe both our theoretical motivation and proposed Top-$k$ algorithm are novel.
>
> Reviewer 8rCb Comment 2: How many runs did you use for the results in Figure 2? Are the conclusions based on statistically significant observations?
>
> Response to Reviewer 8rCb Comment 2: 10 runs were used for each of the plots in Figure 2 and +/- 2 standard error confidence intervals are shown; this is described in the caption for Figure 2. Conclusions are based on statistically significant observations; we generally see that Top-$(k+1)$ outperforms Top-$k$ because their confidence intervals are disjoint (with some exceptions). We note that, as described in the caption for Figure 2, missing points in the plots correspond to settings that did not terminate within a sufficiently large time limit (usually corresponding to the case where $k = d$, i.e., fully optimal decision trees).
>
> Reviewer 8rCb Comment 3: On a related note, in Figure 2, in the case of spambase (and some other cases), MurTree that calculates the optimal decision tree, provides a solution with worse accuracy compared to what the authors achieve with their method. Can the authors shed some light in this?
>
> Response to Reviewer 8rCb Comment 3: MurTree optimizes for train accuracy and, like other ODT algorithms, overfits at higher depths. This is in contrast with our Top-$k$ algorithm, which demonstrates less overfitting for lower values of $k$. Our restriction to the top $k$ features in each recursive step of tree construction can be seen as a form of regularization similar to how Random Forest considers a subset of features at each node split.
>
> Reviewer 8rCb Comment 4: Also, another observation has to do with the fact that on the same graph (spambase), at depth 8, top-2 outperforms top-3. Is there an explanation for this?
>
> Response to Reviewer 8rCb Comment 4: In this setting, Top-$3$ is also overfitting to the training data because there are many features to consider in this dataset. Top-$2$ demonstrates less overfitting and therefore generalizes better.

---

> > ### Comment · Reviewer_8rCb · 2023-08-15
> > **Thank you for the response**
> >
> > Thank you for the response.  I have also read the reviews of others and the response to their comments and I am happy with the responses.  I will upgrade my rating from borderline accept to weak accept.

---

> > > ### Author Response · Authors · 2023-08-18
> > > **Response to Reviewer 8rCb**
> > >
> > > Thank you for your consideration of our rebuttal and the updated score.

---

### Official Review · Reviewer_B1BN · 2023-07-07

**Soundness:** 4 excellent
**Presentation:** 4 excellent
**Contribution:** 4 excellent
**Rating:** 8
**Confidence:** 4

**Summary:**

In the paper authors propose a new algorithm that allows balancing between two approaches to decision tree learning: fully greedy optimization, and exactly optimal decision tree construction.  Authors discuss the long standing line of research that focuses on either ends of this balance, and propose a novel algorithm Top-K that solves the issues of both sides:
- Greedy algorithms are fast, but there are counterexamples that lead to arbitrarily bad accuracy
- Optimal algorithms are inherently slow because of the NP-hardness of the problem.

The Top-K algorithm is backed up by several theorems that show that increasing K allows solving counterexamples that are unsolvable with high enough accuracy using smaller K. While some of the counterexamples involve artificial constructions, authors propose evaluation on real-world datasets that show practical applicability of the proposed method.

**Strengths:**

Most importantly, this is a novel contribution to the field of Decision Trees learning. The proposed algorithm is easy to implement, theoretically sound, and solves major issues of the existing approaches.

On a side note, the paper is well-written, provides convincing motivational examples, a thorough literature review, strong theoretical foundation and a clear description of the algorithms.


**Weaknesses:**

No major weaknesses. There is one concern outlined below, and other questions (which may be addressed here or solved in the future work) are placed in the "Questions" section.
- The paper focuses on a binary classification setting. And features are also binary in the theorems and definitions. This was confirmed in the experimental section, where categorical features have been converted to binary and numerical features had to be thresholded and converted to binary. This raises a question about applicability to non-binary data; and more importantly, to a regression setting (which is equally practically popular, as classification). Clearly stating this limitation or addressing possible ways to extend both theoretical and practical results will help in convincing the reader about usefulness of the proposed algorithm in real world settings and clarify how high is the impact of the paper on the Decision Trees learning field.

**Questions:**

- Since the paper provides one way to balance between fully-greedy and fully-optimal approaches, reader may be curious whether there other possible ways to achieve a balance between those two ends (This may be addressed in the future work, or perhaps authors already have an answer about why Top-K was implemented in a given way, and what could be other possible implementations?)
- There is a line of research on optimizing the Optimal Decision tree algorithm: for example, using Dynamic Programming. While it does not solve the problem of slow running time of the Optimal Algorithm due to NP-hardness of the problem, it allows obtaining different running time bounds, that may be better than fully brute-force search for specific problem sizes and parameters. Would be nice to know whether such optimizations are applicable to the proposed Top-K algorithm (this may be a topic of future work, but perhaps authors already have answers to this question?)
- We see from the experimental results that Top-K outperforms Top-1 on the test set; but in the current implementation Top-K uses binarization and is compared to Top-1 that also uses binarization of the features. Does this binarization significantly reduce the accuracy compared to using a non-binarized dataset? Specifically, what would happen if Top-1 would run on a non-binarized version (because standard Top-1 greedy solvers, such as the ones from scikit-learn do not require binarization of the dataset)?


**Limitations:**

Main concern about limitations is the applicability to problem settings other than binary classification with binary features (outlined in the above sections of the review)

---

> ### Author Rebuttal · Authors · 2023-08-09
>
> We thank Reviewer B1BN for their insightful comments and helpful feedback.
>
> Reviewer B1BN Comment 1: The paper focuses on a binary classification setting. And features are also binary in the theorems and definitions. This was confirmed in the experimental section, where categorical features have been converted to binary and numerical features had to be thresholded and converted to binary. This raises a question about applicability to non-binary data; and more importantly, to a regression setting (which is equally practically popular, as classification). Clearly stating this limitation or addressing possible ways to extend both theoretical and practical results will help in convincing the reader about usefulness of the proposed algorithm in real world settings and clarify how high is the impact of the paper on the Decision Trees learning field.
>
>
> Response to Reviewer B1BN Comment 1: We would like to clarify a few points regarding binarization of the datasets. First, we chose to consider binary datasets primarily for comparison with prior work, as many SOTA Optimal Decision Tree algorithms are restricted to binary datasets, and for theoretical analysis.
>
> However, this is not a restriction of our work as our proposed Top-$k$ algorithm can be easily generalized to non-binary data. For example, we may also consider the $k$ best feature splits by considering the top $k$ (feature, threshold) pairs or, instead, consider the top $k$ features and select the best threshold for each feature based on a heuristic – either approach would work in the non-binary data setting.
>
> Lastly, we note that every dataset can be appropriately binarized by appropriately thresholding each feature, and an advantage of our proposed Top-$k$ algorithm is that it works better than existing approaches when this binarization process introduces many more features.
>
> The modification for regression is also simple. In classification, at each leaf, we put the most frequent label among training points that reach that leaf. In regression, we would simply put the mean of the label among such training points (e.g., if the goal is to minimize $L_2$ loss).
>
> Reviewer B1BN Comment 2: Since the paper provides one way to balance between fully-greedy and fully-optimal approaches, reader may be curious whether there other possible ways to achieve a balance between those two ends (This may be addressed in the future work, or perhaps authors already have an answer about why Top-K was implemented in a given way, and what could be other possible implementations?)
>
> Response to Reviewer B1BN Comment 2: To the best of our knowledge, Top-$k$’s ability to interpolate between fully-greedy and fully-optimal approaches is novel. Indeed, this is a key strength of our work.
>
> We note that a recent paper [1] extends DL8.5 to achieve better anytime behavior via iterated limited discrepancy search. This approach is much slower than Top-$k$, because the number of iterations to achieve optimality is very large, and there are no theoretical performance guarantees for intermediate steps of the algorithm, but this work may nonetheless be of interest to the reviewer. We will add [1] to our related work section.
>
> [1: ​​Time Constrained DL8.5 Using Limited Discrepancy Search, ECML PKDD 2022]
>
> Reviewer B1BN Comment 3: There is a line of research on optimizing the Optimal Decision tree algorithm: for example, using Dynamic Programming. While it does not solve the problem of slow running time of the Optimal Algorithm due to NP-hardness of the problem, it allows obtaining different running time bounds, that may be better than fully brute-force search for specific problem sizes and parameters. Would be nice to know whether such optimizations are applicable to the proposed Top-K algorithm (this may be a topic of future work, but perhaps authors already have answers to this question?)
>
> Response to Reviewer B1BN Comment 3: These optimizations are indeed applicable to our proposed Top-$k$ algorithm and we have already implemented them. Appendix Figure 5 of our original submission describes our algorithm with these optimizations. Our implementation is actually an extension of DL8.5, a state-of-the-art Optimal Decision Tree (ODT) algorithm, and incorporates many of its optimizations.
>
> Reviewer B1BN Comment 4: We see from the experimental results that Top-K outperforms Top-1 on the test set; but in the current implementation Top-K uses binarization and is compared to Top-1 that also uses binarization of the features. Does this binarization significantly reduce the accuracy compared to using a non-binarized dataset? Specifically, what would happen if Top-1 would run on a non-binarized version (because standard Top-1 greedy solvers, such as the ones from scikit-learn do not require binarization of the dataset)?
>
> Response to Reviewer B1BN Comment 4: We would like to emphasize that we used binarization primarily for comparison with prior SOTA Optimal Decision Tree algorithms, which require binary datasets. We anticipate that extending Top-$k$ to the non-binary case would be simple and the comparisons to CART demonstrate similar results (see our response to Comment 1).

---

> > ### Comment · Reviewer_B1BN · 2023-08-17
> >
> > I would like to thank the Authors for their detailed response, it answers my questions and agrees with my initial evaluation score.

---

> > > ### Author Response · Authors · 2023-08-18
> > > **Response to Official Comment by Reviewer B1BN**
> > >
> > > Thank you for taking the time to review our paper and rebuttal.

---

### Official Review · Reviewer_UM1K · 2023-07-07

**Soundness:** 3 good
**Presentation:** 4 excellent
**Contribution:** 3 good
**Rating:** 7
**Confidence:** 4

**Summary:**

The paper presents a novel generalization of classic greedy decision tree algorithms called Top-k, which considers recursively the top k attributes for possible splits rather than the single best attribute. It enhances traditional decision tree learning algorithms such as ID3, C4.5, and CART, and is expected to be a significant improvement on these established methodologies. The authors provide both theoretical and empirical proof supporting the effectiveness of this modification.

Theoretically, the authors establish the greediness hierarchy theorem that demonstrates there exist data distributions for which one can expect significant improvements in accuracy as k increases. Empirically, they test Top-k against standard greedy algorithms and optimal decision tree algorithms using real-world datasets. The experimental results reveal that for the same tree depths, Top-k significantly outperforms the classic greedy methods while also being much more scalable than optimal decision tree algorithms.

**Strengths:**

- The paper makes a novel contribution to the field of decision tree learning by introducing the Top-k generalization of classic greedy algorithms. The concept is simple and intuitive, yet it has a profound effect on the performance of the algorithm.

- The authors have provided a detailed theoretical basis for their work. The introduction of the greediness hierarchy theorem gives clear theoretical evidence of the potential benefits of considering more attributes for splits. This robust theoretical grounding is a key strength of this paper.

- A comprehensive experimental evaluation is carried out to back the theoretical findings. The authors evaluate the Top-k approach using UCI datasets, comparing the results with both classic greedy algorithms and optimal decision tree algorithms. The performance of Top-k is compelling, exhibiting significant improvements over classic methods and demonstrating comparable scalability.


**Weaknesses:**

- While Top-k demonstrates stronger performance than classic greedy algorithms for the same tree depths, an additional, more fair setup would be comparing the test accuracy achieved in comparable training time.
- The runtime scales exponentially with the depth h.


**Questions:**

- In gradient-boosting tree ensembles, the depth of each tree is usually small. Can Top-k improve the predictive performance of GBDTs?

**Limitations:**

The limitations have been discussed in the Conclusion section.

---

> ### Author Rebuttal · Authors · 2023-08-09
>
> We thank Reviewer UM1K for their insightful comments and helpful feedback.
>
> Reviewer UM1K Comment 1: While Top-k demonstrates stronger performance than classic greedy algorithms for the same tree depths, an additional, more fair setup would be comparing the test accuracy achieved in comparable training time.
>
> Response to Reviewer UM1K Comment 1: We would like to clarify the motivation for our proposed algorithm. Optimal Decision Trees (ODTs) are popular because they can be generated in “reasonable time” for shallow depths and offer better accuracy than greedy approaches, not because they achieve better balance between speed and accuracy. Our proposed algorithm, Top-$k$, expands the range of datasets and depths that can be considered in “reasonable time” while still improving upon the accuracy of greedy approaches. As such, our experiments set a “reasonable” bound on the runtime (10 minutes) and take accuracy measurements under this restriction, because accuracy is our key metric (not training time).
>
>
> Reviewer UM1K Comment 2: The runtime scales exponentially with the depth h.
>
> Response to Reviewer UM1K Comment 2: We also note that the runtime scales exponentially with the depth $h$ for all optimal decision tree algorithms (where $k = d$), and for CART and other greedy algorithms (where $k = 1$), where there are exponentially many internal nodes. In future work, we hope to improve upon $(2k)^h$ naive bound as $k$ increases using the techniques from the ODT literature.
>
> Reviewer UM1K Comment 3: In gradient-boosting tree ensembles, the depth of each tree is usually small. Can Top-k improve the predictive performance of GBDTs?
>
> Response to Reviewer UM1K Comment 3: This is an exciting direction for future work. Research into optimal decision tree ensembles is relatively unexplored. The original MurTree paper compares MurTree ensembles to Random Forests; however, further systematic study is necessary. We note that in our proposed algorithm, Top-$k$, the $k=1$ case corresponds to greedy algorithms and therefore subsumes greedy gradient-boosting ensemble methods (with the appropriate training objective). We suspect our method could be applied with $k > 1$ to improve upon existing greedy gradient boosting ensemble methods, though we leave a thorough exploration of this to future work.

---

> > ### Comment · Reviewer_UM1K · 2023-08-20
> >
> > Thanks for your clarification. I will keep the score the same.

---

> > > ### Author Response · Authors · 2023-08-21
> > > **Response to Reviewer UM1K**
> > >
> > > Thank you for taking the time to review our paper and rebuttal.

---

### Official Review · Reviewer_mdbV · 2023-07-21

**Soundness:** 3 good
**Presentation:** 2 fair
**Contribution:** 3 good
**Rating:** 3
**Confidence:** 5

**Summary:**

The paper studies an extension of greedy decision tree algorithms. The main idea is, instead of simply greedily selecting the best feature, to consider several features. This can also be seen as a restriction of optimal decision tree algorithms, so in some sense the studied approach lies somewhere in between greedy and exhaustive algorithm. The point is that greedy approach are fast, optimal algorithms are slow, so something in between may be desirable.

The authors provide theoretical results and some experiments to illustrate a few points.

Even though the idea of selecting a subset of the features is emphasized in the paper, I find this questionable, e.g., I find that a minor and rather natural point. In my view the main contribution are the theoretical results - these are novel to the best of my knowledge and provide a nice way to formalize relatively intuitive ideas. I think the paper would benefit from focusing on this theoretical aspect since this is a unique point.

While I am enthusiastic about the theoretical results, I feel that there are many (small) issues with the paper that need to be addressed before publication. In particular, the relationship with related work needs to be expanded and discussed in more depth, the experiments go in the right direction but the claims are not convincingly backed up by data, and there are several (minor) technical issues.


**Strengths:**

The theoretical results sound relevant. Recent recent tree works concerning "practical" algorithms do not provide virtually any theoretical results, and in this work this is studied formally. The authors also point of the artificialness of the theoretical results, and then propose a more "realistic" mathematical setting. All of these points are welcome contributions, and highlight (potential) for good technical depth.


**Weaknesses:**

Conceptual

The related work is treated very briefly: the related work section is half a page. The text gives a bird's eye overview of the field, which is good, but even this text does not relate to the work done in the paper. I would like to see some more discussion on how related algorithms work and have the authors draw the connection to this approach. This places the work in the right context which is important when publishing a paper. This absence of direct relation to related work is also reflected in the paper later on.

Since the paper covers theory and the authors consider a parameter k, it would be natural to relate this work to the paramaterized complexity of decision trees paper [Parameterized Complexity of Small Decision Tree Learning, AAAI 2021]. This would provide more rigorous analysis.

The text presents the results as surprising, as going from k to k+1 may make a large difference even though it is only an increase of one. This is a nice observation, and should be mentioned, but the text needs to be toned down. Since the number of decision trees grows exponentially with the number of features, it is clear that going from k to k+1 leads to an exponential increase in number of trees. So it makes sense that this has an impact on accuracy. Note that this does not diminish the value of the contribution, it merely puts it in the right perspective.

The authors state that considering "top-k" compared to "top-1" introduces a "mild" increase in complexity. However the algorithm increases _exponentially_ with the k parameter, which is also shown in this paper (Claim 3.1). It could be that in practice, if the k is small, the impact is not large in terms of absolute nubmer of seconds, but in theory this is still a large complexity increase. Setting k = number of features leads to the exhaustive algorithm, which is exponential.

I find the idea of top-k interesting and relevant but the idea is rather small. The value of the paper in my view is in the theory part and (possibly improved) experiments. This could be reflected more in the text. Note that the top-k approach has already been implemented in MurTree but I am not sure if there have been thorough evaluations.

Definition 3 and Lemma 3.2: these follow directly from the algorithm (Figure 1), so it is odd to pack the results as a lemma that has proof in Appendix, makes it sounds more complicated that it is. A simpler proof, if needed to be included, is to show by induction on the recursively formulation in Figure 1.

Parallelization is mentioned as an "easy" task. First, this is the case with virtually every decision tree algorithm. Second, the situation is only easy on the surface level: it is not trivial to parallelize techniques used by optimal decision tree algorithms (bounding, caching) that give significant runtime improvements, so a trivial parallelization can leads to degradation of performance.

Experiments

The experiments mention that 100 datasets are available, but the authors select a random subset of 20. But why only 20, why not all? Why not take available datasets that have been directly used in previous works? This seems odd given the ability of the datasets, and the runtimes are still managable.

The conclusions are made on cherry picked datasets, i.e., only a few datasets even though many datasets are available. The scalability experiments shows that top-k scales much better: this is obvious based on the complexity, but it is still interesting to see. However the dataset seems to be extreme: while other considered datasets have 50-300 features, this dataset has 1400 features. While it is a good idea to use an extreme example to emphasize the scalability problem, this needs to be done all datasets and not just on one.

The comparison with MurTree and GOSDT is also not entirely proper, since the algorithms solve slightly different problem. MurTree is the most similar, GOSDT is slightly different because of the alpha parameter and no explicit depth. This would need to be discussed, and/or setting the same alpha parameter for MurTree so that the comparisons are more aligned (although in this case this paper does not support the alpha parameter, the authors could also include this parameter).

The conclusion that after some k value there are no gains in accuracy: this is intuitive and nevertheless interesting, but the conclusion is made on three datasets. It is needed to do more thorough experiments on all datasets to make this claim.

The authors discuss that top-k can sometimes even lead to better test accuracy. This is problematic for two reasons. First, the method of comparison (dl8.5 in this case) does not optimise in any way for test accuracy, so the comparison is odd. Second, the authors compare against full decision trees, which are prone to overfitting. Since the algorithm is incentivised to use all nodes, it will use all nodes and capture noise in the training data, so it is expected that it will not generalize well. The fix here is to do tuning with not only the depth, but the number of nodes the algorithm is allowed to use, or alternatively use the alpha parameter.

Minor

Figure 5 seems important but it is placed in the Appendix.

Please provide the intuition behind the proofs. This would add more depth to the paper.

Since monotone is an important concept, please introduce monotonicity for completeness.

**Questions:**

I believe the paper does have potential but there are many issues as outlined in the Weakness section. Since there are many issues, I do not have specific questions at this point. I hope my review will help the authors in revising the work, I would be happy to see it refined and published later on. In case I misunderstood something the authors are welcome to point out flaws in my reasoning.


**Limitations:**

This is somewhat discussed in the paper.

---

> ### Author Rebuttal · Authors · 2023-08-10
>
> We thank Reviewer mdbV for their helpful feedback. We provide a point-by-point response to Reviewer mdbV’s comments below.
>
> Conceptual
>
> W1: We have synthesized a body of additional related work and would be happy to provide it in an additional “Official Comment” if requested. We have currently omitted it from our Rebuttal to respect the length limit.
>
> W2: We will cite this paper. While their work considers a bound on feature domain size, Top-k considers all possible features, but only branches on the first k in each recursive step based on a greedy heuristic.
>
> W3: We will make the text more muted.
>
> W4: We agree with this point. We meant to describe a small absolute increase in runtime and will make this more clear.
>
> W5: We would like to clarify a potentially critical misunderstanding: our proposed algorithm, Top-k, does not select a fixed or predetermined subset of features. Instead, Top-k considers all possible features at each step but only explores the top k of them, as determined by greedy heuristics. Top-k is not doing feature subset selection and, rather, is more similar to a beam search.
>
> MurTree has not implemented the Top-k approach and only considers maximum depth, sparsity, and maximum number of internal nodes. In a comparison to Random Forest done in the MurTree paper, MurTree is run on several randomly generated subsets of the feature domain. In Top-k, all features are considered and k refers to the number of features that are chosen to expand in each step, as ranked by a greedy heuristic. To our knowledge, Top-k has not yet been incorporated into any ODT algorithms.
>
> W6: We will move the proof to the main paper.
>
> W7: Reviewer mdbV is correct about parallelization; we will correct this.
>
> Experiments
>
> W8: The subset on which we ran experiments were a true random sample, not cherry-picked (as described on Lines 269-270). The only dataset that was chosen for illustrative purposes was the dataset with 1400 features. Running on the random subset took over 8 days. Unfortunately, running all 100 datasets would be prohibitively expensive. This is because DL8.5, MurTree, and GOST required large memory allocations and took over 10 minutes to train on most datasets.
>
> W9: Apart from the dataset with 1400 features, which was chosen for direct comparison with the GOSDT paper, all other datasets were a true random sample of the available datasets. This is described on Lines 269-270 of the original submission.
>
> W10: In our experiments, we used the default hyperparameters for the baseline algorithms. Top-k and MurTree share the same depth constraint. For GOSDT, we set the regularization coefficient to 2^{-depth}, as described on Line 302 of our submission. This choice of regularization coefficient sets the same hard depth constraint implicitly. Even in this setting, GOSDT exhibits significantly worse runtime compared to Top-k.
>
> W11: In an additional experiment presented as Figure 1 in the “global” comment to all reviewers, we run this experiment with a 10 second time limit per run, letting Top-k return the best tree discovered within the time limit. We observe that the same trend holds in many other datasets: that increasing k beyond a point does not result in additional gains in accuracy.
>
> In Figure 2 of our additional experiments, we also run a similar experiment as done for Figure 4. We find that the pattern observed by Figure 4 of our original submission still holds: that increasing k provides marginal or no benefit beyond a certain point and can even lead to overfitting at higher depths.
>
> W12: We agree with Reviewer mdbV’s observation that the baseline ODT algorithms overfit and do not generalize well at higher depths.
>
> We also note that the restriction on depth implicitly constrains the number of nodes in a tree. Even when ODT algorithms are not able to expand fully due to this depth constraint, we observe that they still overfit.
>
> We ran an additional experiment to provide further empirical evidence of this observation. In this experiment we replicated Figure 2 from the original paper but expanded to higher depths and values of k by imposing a time limit of $10$ seconds and having Top-k return the best tree discovered within the time limit. The results of this experiment are displayed in Figure 1 of our additional experiments, provided in a “global” comment to all reviewers. We observe that increasing k provides marginal benefits at low to mid depths, and at higher depths, higher values of k experience severe dips in performance. This is due to overfitting and an inability to fully explore the search space.
>
> For lower k values, Top-k is able to expand to much higher depths and still perform well, whereas ODT overfits. Indeed, this is a weakness of ODT algorithms and a strength of our proposed Top-k approach.
>
> We also note that Optimal Decision Trees often cannot be trained at very high depths because their training time and memory requirements are too large.
>
> Minor
>
> W13: We will move Figure 5 to the main paper.
>
> W14: The intuition behind Theorems 2 and 3 is that we can design data sets in which the features are in two disjoint groups, “truly important” and “seemingly important” where:
> The “seemingly important” features are rated more important by the feature scoring function than the “truly important” features,
> Any decision tree built from only “seemingly important” features will have low accuracy.
> It is possible to build a decision tree using only the “truly important” features that achieves high accuracy.
>
> To prove our result, we set the number of “seemingly important” features so that Top-k will only select “seemingly important” features, but Top-$(k+1)$ can see past the “seemingly important” features to select the “truly important” features. This construction is possible because impurity heuristics only measure first order correlations; whereas the truly important features have higher order correlations.
>
> W15: We define monotonicity on Line 246 of our original submission.

---

> > ### Comment · Reviewer_mdbV · 2023-08-15
> >
> > W1 (related work): I am taking this as you agree that the related work does not provide the connection with other works to a sufficient extent. My main concern is because there is a lack of in-depth understanding of related work, there are a number of issues that propagate in the paper as outlined in the review. This is a serious issue. Nevertheless since you already produced a more thorough treatment of your work with respect to the literature, it would be interesting to see.
> >
> > W2 (parametrized complexity): the point is that the paper also considers a case that limits the number of features. This is different from your setting, but related enough to be investigated given that you also provide a theoretical contribution.
> >
> > W5 (top-k): your approach comes across well in the paper, it is clear: at each step you order the features according to their score and consider only the first k features. It is a nice way to scale optimal algorithms. You could ignore that comment of mine, the approach seems to be implemented in the code but there is no thorough evaluation or discussion in the paper, so my comment can be ignored.
> >
> > W8 (cherry-picked): the point still hold: selecting 20 out of 100 datasets seems odd. I am not entirely about your justification. Running 100 datasets with 10 minutes each would take less than a day per algorithm. With three algorithms to compare, that is three days. On all datasets, assuming that the maximum time allocated is used. Considering some form of k-fold validation could be used, this could increase the experimental runtime, but it still seems reasonable, especially if a computing cluster is used.
> >
> > W8 (follow up): are these same datasets used in other decision tree works? For instance, DL8.5 experiments with datasets with binary datasets, do you have an idea how your datasets relate to those? This could be clarified in the paper.
> >
> > W8 (dataset with 1400 features): this is a nice example for demonstrating the issue with a lot of features. But the point is that this is an extreme case dataset, it seems that most of your datasets have 300 features or so, which seems to be well within reach of the other algorithms. What happens when considering datasets with 300 or less features?
> >
> > W10 (hyper-parameter tuning): there seems to be a misunderstanding. I am not asking about hyper-parameter tuning. My comment was that comparisons should be done apples-to-apples, i.e., the algorithms should be solving the exactly same problem if you want to make claims about the algorithms.
> >
> > Note that setting the regularization coefficient to 2^{depth} is not the same as limiting the depth. There is a difference when you penalize for using each node and simply saying that any number of nodes is okay until a given depth. The regularization does bound the depth, but the objective is different, since the approach pays a cost for every node, so it is not the same problem anymore.
> >
> > W12 (over-fitting): I am not sure the authors understood my comment here. The comparison does is not appropriate, but there is no acknowledgement from the author side about this. The point is that the model that DL8.5 produces is not good for generalization, because it _only considers full trees_.
> >
> > Comparing and saying our approach is better on the out-of-sample accuracy is odd, because DL8.5 does not attempt to compete on out of sample accuracy. A better way would be to tune the number of nodes the optimal decision tree can use on a per dataset basis.
> >
> > Adding limits on the depth of course limits the number of nodes, but this difference can be significant: a depth four tree can have anywhere from 0 to 31 nodes (including leaf nodes). That is a large difference, and an optimal decision tree algorithms that is free to use as many nodes as possible will likely end up using all of them, even if this captures some noise, because that is the objective that is optimized (no consideration for out of sample).
> >
> > If the ODT algorithms cannot reach the maximum depth, and seemingly use less nodes, this is not appropriate as an example for comparison, because of how the algorithms work. The anytime performance is not considered, so stopping the algorithms before the timeout does not necessary produce a "good" tree. This is due to the way the search space is explored in the algorithm.
> >
> > W? (increasing k): I would expect that indeed increasing k does not lead to benefits after some point, and it is nice to experimentally show.
> >
> > Note that in general it is not necessarily about showing better numbers, but showing an appropriate discussion analyzing why those numbers are the way the are. Simply saying this approach performs worse, here are experiments, is not appropriate since the circumstancs of the comparison and the methods need to be understood well.
> >
> > Overall I think the paper has potential, but as evidenced by lengthy discussion, there are many points that need to be carefully considered.

---

> > > ### Author Response · Authors · 2023-08-18
> > > **Response to Official Comment by Reviewer mdbV (1 of 2)**
> > >
> > > We thank Reviewer mdbV for their detailed feedback and time considering our paper and rebuttal.
> > >
> > > W1: We agree that our related work could be improved and provide our updated related work below.
> > >
> > > **Optimal Decision Tree (ODT) Search [Extended]:**
> > > In light of drastic improvements in the efficiency of modern solvers, various formulations of the optimal decision problem have been proposed in recent years (Mixed Integer Programming [5] / Binary Linear Programming [6] / Constraint Programming [7] / SAT [8]).
> > > Several dynamic programming algorithms (e.g. DL8.5 [4], MurTree [3]) have also been developed for the task. These algorithms use specialized data structures, pruning, and subtree-decomposition of the search space to achieve better performance than the former, more general approaches.
> > > Our implementation of Top-$k$ is an extension of the DL8.5 library and so benefits from many of its optimizations.
> > > **Optimal Sparse Decision Tree (OSDT) Search:**
> > > A natural per-leaf sparsity penalty was also recently incorporated in order to regularize decision trees. This penalty was first proposed in OSDT [9] and GOSDT [2] which discover OSDTs via sparsity-based bounds and a priority-queue-based dynamic programming algorithm. While popular, this form of regularization appears somewhat arbitrary, comes with no theoretical guarantees, and does not appear to generalize well in comparison to ODT or even greedy algorithms. In comparison, Top-$k$ can be viewed as a form of regularization for which we provide solid theoretical motivations backed by empirical evidence.
> > > **Between Greedy and Optimal Decision Trees:**
> > > There have been many semi-greedy approaches proposed to construct binary classification trees. We restrict our focus to approaches which encapsulate both greedy tree construction algorithms and optimal decision tree algorithms.
> > > A very common approach that fits into this category is lookahead: looking several splits into the future in each recursive step of decision tree construction. To the best of our knowledge, lookahead is empirically rather than theoretically motivated, and early work on lookahead in decision tree induction suggested that it may actually produce worse trees [10]. However, it has found some empirical success, especially in the anytime setting [11, 12, 13].
> > > As with ODTs, however, lookahead scales poorly on datasets with high feature counts. Top-$k$ can be viewed as an alternative semi-greedy approach that restricts the width of the decision tree search rather than the height, making it invariant to feature count and potentially more scalable as a result.
> > >
> > > [3: MurTree: Optimal Classification Trees via Dynamic Programming and Search, JMLR 2022]
> > >
> > > [4: Learning Optimal Decision Trees Using Caching Branch-and-Bound Search, AAAI 2020]
> > >
> > > [5: Optimal classification trees, Machine Learning 2017]
> > >
> > > [6: Learning Optimal Classification Trees Using a Binary Linear Program Formulation, AAAI 2019]
> > >
> > > [7: Learning Optimal Decision Trees using Constraint Programming, IJCAI 2020]
> > >
> > > [8: Learning optimal decision trees with sat, IJCAI 2018]
> > >
> > > [9: Optimal Sparse Decision Trees, NeurIPS 2019]
> > >
> > > [10: Lookahead and Pathology in Decision Tree Induction, IJCAI 1995]
> > >
> > > [11: Lookahead-based algorithms for anytime induction of decision trees, ICML 2004]
> > >
> > > [12: Any time induction of decision trees: an iterative improvement approach, AAAI 2006]
> > >
> > > [13: Look-Ahead Mechanism Integration in Decision Tree Induction Models, Advanced in Web Intelligence and Data Mining 2006]
> > >
> > >
> > > W2: We will draw a detailed comparison to this work in our related work section. Specifically, we will emphasize that our trees are not restricted to a subset of the features but rather considers all features, whereas the reference work considers a subset of features. We also note that the referenced work does not demonstrate the performance of trees with restricted feature spaces, and that their parameter that governs the number of selected features is only one of several parameters in their paper, so is not the focus of their work.
> > >
> > > W5: We acknowledge that Reviewer mdbV has instructed us to ignore the comment that top-k is implemented in MurTree. For completeness, however, we do not believe that it has been implemented in their code. Could Reviewer mdbV point us to the place they believe it has been implemented in the code? We are looking at https://bitbucket.org/EmirD/murtree/src/master/main.cpp.
> > >
> > > We believe the lack of existing Top-k implementations is important to the novelty of our approach.

---

> > > > ### Author Response · Authors · 2023-08-18
> > > > **Response to Official Comment by Reviewer mdbV (2 of 2)**
> > > >
> > > > W8 (cherry-picked and dataset with 1400 features): In the reviewer’s calculations that 100 datasets with 10 minutes each will take less than a day per algorithm, they are neglecting the many different hyperparameter settings for each dataset. We meant to say that it takes a maximum of 10 minutes per setting for each of at least (6 values of $k$)*(6 depths)*(10 train/test splits) = 360 hyperparameter settings.
> > > >
> > > > For the accuracy comparison for varying k values and depths, it took approximately (6 values of $k$)$\times$(6 depths)$\times$(10 train/test splits)$\times$(20 datasets)$\times$(10 mins/run) = 50 days to run on all 20 datasets with a time limit of 10 minutes. For the variation in k values just on the 20 datasets, for which there were a total of 2118 features, it would have taken approximately (2118 k values)(10 train/test splits)(10 min) = 147 days. For the speed datasets, with seven (not three) algorithms (GOSDT, MurTree, Top-1, Top-2, Top-4, Top-8, Top-16), it would have taken approximately
> > > > (7 algorithms)(30 feature subsets + 30 sample subsets)(20 datasets)(10 min) = 58 days.  Furthermore, it is difficult to parallelize experiments due to the massive memory consumption of many baseline algorithms.
> > > >
> > > > We prioritized re-running the first two experiments on all datasets for the rebuttal (with a 10 second time limit) because it seemed that the reviewer had agreed that the search space explored by Top-k was smaller and so the scaling we had already observed in the third experiment was to be expected.
> > > >
> > > > W10: We thank the reviewer for this comment. We agree that setting the regularization coefficient to 2^{depth} is not equivalent to limiting the depth because the regularization penalties behave differently. We were trying to say that setting the regularization coefficient to 2^{depth} also implicitly sets a maximum depth.
> > > >
> > > > However, we believe that incorporating additional objectives into Top-$k$ is outside the scope of this paper. We only claim to scale better than MurTree and GOSDT. As noted previously, the search space explored by Top-$k$ is much smaller than that explored by ODT algorithms due to restriction on $k$, so we expect it to scale better.
> > > >
> > > > Furthermore, MurTree and Top-$k$ use the same parameters in our experiments (same maximum depth). GOSDT is generally known to be quite slow, which is consistent with our experiments. (In GOSDT, the sparsity objective is subtree-decomposable meaning it does not make the problem any more difficult. In fact, the search space explored by GOSDT with a sparsity penalty of 2^{-maxdepth} is much smaller than the one explored by MurTree and Top-k).
> > > >
> > > > W12: We believe that we are actually saying the same things as Reviewer mdbV here. We agree that models like DL8.5 generally consider full trees to fit the training data, and therefore often identify spurious correlations and do not generalize well (i.e., overfit). This is exactly a shortcoming of DL8.5 and other ODT algorithms that we are trying to highlight. Furthermore, the poor anytime behavior of ODT algorithms and DL8.5 due to how the search space is explored is another shortcoming of those approaches; our proposed method has better anytime behavior.
> > > >
> > > > We consider improving the baseline algorithms to be outside the scope of our contribution; instead, we compare our proposed method to current state-of-the-art implementations. ODT algorithms like DL8.5 claim to generalize better than greedily constructed trees. We show in this paper that Top-$k$ can improve the performance of ODT algorithms which constrain depth, the most popular version of ODTs.

---

> > > > > ### Comment · Reviewer_mdbV · 2023-08-21
> > > > >
> > > > > In case you are drawing conclusions based on experiments, it is important the experimentation is extensive enough. Otherwise you will leave room for doubt. In your concrete case, using 20% of the datasets to make conclusions raises an obvious question, why only 20%? Would the results be differently if all datasets would be used? What if datasets from other papers are considered? Etc. This is similar across any discipline that relies on experimental results, and machine learning definitely fits into this category. Generalizing conclusions from a limited sample size is bad practice.
> > > > >
> > > > > There are many ways around this issue. The most straight-forward approach is to run experiments on a computational cluster, and there you would get all the results within a day. If such hardware if not directly available to you, you can rent out a cluster, and this will cost only a fraction of the cost it would to attend NeurIPS. If for some reason a cluster is out of your reach, then more carefully analysis on hyper-parameters is needed. It is clear that not all algorithms will make use of the full timeout on all parameters. You can also limit the grid search in some reasonable way that does not hurt your story. There are many possibilities here, but the bottom line is that running comprehensive experiments is standard.
> > > > >
> > > > > If you want to make a direct comparison with a method that uses a sparsity coefficient, then you need to be able to optimize the sparsity coefficient as well. Otherwise the comparison is not direct and you are not comparing algorithms but something else (models?). It could be that you do not need a direct comparison. You need to evaluate whether that is what you need. But your choice needs to be well justified in the text and clearly stated.
> > > > >
> > > > > The issue you are highlighting about full trees not generalizing is straight-forward, but in the paper it is presented as a discovery. You are comparing to an algorithm (DL8.5) that has _no consideration for out of sample accuracy_, and claiming you are doing better. In the DL8.5 paper (AAAI 2020), there is not even an out of sample comparison in the paper, everything is done on the whole dataset / training set. The contribution is on the algorithm to explore the search space but there no tips on how this would apply in practice, and at best the claim is a reference to other papers like Bertsimas and Dunn 2017 where they show the value of optimal decision trees. Concretely with the case of comparing to DL8.5 on out of sample, this is something you would do the experiments on just to check if everything is as expected with your method (as a test, to check if there are errors with your code), place the results in the Appendix, and write two-three sentences in the paper saying that well full trees do not generalize well, and use the remaining space to add more discussion on things that are new. The summary is that the comparison needs to be informative, non-misleading, and fair.
> > > > >
> > > > > Some final comments on the related work.
> > > > >
> > > > > "A natural per-leaf sparsity penalty was also recently incorporated in order to regularize decision trees. This penalty was first proposed in OSDT [9] and GOSDT [2]"
> > > > >
> > > > > This is not correct. The sparsity penalty is a text book idea that dates way back. In the context of optimal decision trees, Bertsimas and Dunn 2017 (and possibly Verwer and Zhang 2017) used this before the referenced papers.
> > > > >
> > > > > Reference 7 is not appropriate. You are referencing the sister conference track paper at IJCAI. This is a short paper that is not meant to be cited, it contain a very short summary of the work. Instead you should reference the original paper on which that sister track paper is based on.
> > > > >
> > > > > "While popular, this form of regularization appears somewhat arbitrary, comes with no theoretical guarantees, and does not appear to generalize well in comparison to ODT or even greedy algorithms"
> > > > >
> > > > > This seems unjustified, e.g., there is no reference or credible source for this claim. You can have a look at the Bertsimas and Dunn 2017 paper on how they tune the sparsity coefficient, and they seem to have success with it.
> > > > >
> > > > > "Top- can be viewed as a form of regularization for which we provide solid theoretical motivation"
> > > > >
> > > > > You could clarify how your theoretical results provide hints on how top-k should be applied in practice for regularization, e.g., you could highlight the theoretical contribution that shows that setting k too high is "bad" which is what I would expect from a regularization technique. I think this is not what you do, but from the text that is what I would think you do (I do admit now that I did not go to double check the paper, so I may be wrong).
> > > > >
> > > > > In related work you could also give the readers the idea how the most related algorithms work, and how your approach relates to those.
> > > > >
> > > > > Minor comment, there are other SAT papers on decision trees. You can easily find them by seeing which papers reference that paper.
> > > > >
> > > > > Very minor, 'drastic' generally implies negative connotations.

---

> > > > > > ### Comment · Reviewer_mdbV · 2023-08-21
> > > > > >
> > > > > > Thank you for the discussion so far, I will take this further with other reviewers in the post rebuttal period.

---

> > > > > > > ### Author Response · Authors · 2023-08-21
> > > > > > > **Response to Reviewer mdbV**
> > > > > > >
> > > > > > > Thank you for the additional discussion. Unfortunately, we will not have time to respond to all of Reviewer mdbV's points before the author-reviewer discussion period closes, but we appreciate Reviewer mdbV's feedback.

---

### Author Rebuttal · Authors · 2023-08-09

Attached please find new experimental results which we refer to in our rebuttals to reviewers.

---

### Decision · Program_Chairs · 2023-09-21

**Decision:**

Accept (poster)

**Comment:**

The paper extends decision tree construction from greedy to beam search, ie. allowing to optimize recursively over k splits at each node instead of 1. When k equals to 1, we have standard greedy trees and when k equals to the number of features, we have optimal decision trees. They show theoretically that there exist problems (kinds of XOR problems) for which increasing k by 1 is beneficial. They then perform experiments in which they compare their method to standard greedy trees and optimal decision tree algorithms on 20 problems.

One of the reviewers strongly suggests rejection, while all other reviewers clearly recommend acceptance. The main limitations of the work that were identified in the reviews and subsequent discussions are the following:
- The treatment of the related literature is not complete enough (reviewer mdbV spotted several missing references and errors in the treatment of related works).
- The paper overlooks a bit the computational complexity of the proposed method. The Top-k method is more efficient than existing optimal tree algorithms for small k but it scales exponentially with respect to tree depth and is probably not competitive with these methods when k is at its maximum value. The argument that it can be made trivially efficient by parallelisation is a bit naive, as this would require an exponential number of jobs and this would also benefit other methods. The reviewers and I suggest to remove or tone down the discussions about parallelisation.
- Experiments have limitations. There were discussions about the selection of only 20 datasets and the fairness of the comparison against optimal tree algorithms. The three positive reviewers are however convinced that the datasets were not cherry-picked and that the experiments provide enough evidence of the interest of the approach. I tend to agree although I believe that some important experiments are nevertheless missing. It would have been more informative to compare the performance of top-k against optimal tree algorithms on the training set instead of the test set (since this is what optimal tree algorithms are targeting, as highlighted by reviewer mdbV).  Additionally, the experiments do not clearly highlight the benefit of top-k against standard greedy trees. Indeed, the experimental setting is very far from the way one would use decision trees in practice (mostly the features would be not be discretised and the tree complexity would be tuned). It’s not obvious that top-k would still be performing well in a more realistic setting. Actually, given the way the results are presented in the paper, I found it difficult to really compare top-1 with top-k, in terms of test accuracy and computing times (it's not easy to link improvements in Figure 2 with computing times in Figure 6).

Despite these limitations, all reviewers however agree that the proposed method is novel, relevant, and that it deserves to be studied. They are also positive about the theoretical analyses in the paper. For these reasons, I recommend acceptance. However, I urge the authors to consider Reviewer mdbV's comments about the treatment of related works and also to try to address the other points mentioned in the reviews and this meta-review, in order to strengthen their contribution.